# Nucleotide binding to the ATP-cone in anaerobic ribonucleotide reductases allosterically regulates activity by modulating substrate binding

Ornella Bimai[1], Ipsita Banerjee[2†], Inna Rozman Grinberg[1], Ping Huang[3], Lucas Hultgren[4], Simon Ekström[4], Daniel Lundin[1], Britt-Marie Sjöberg[1], Derek T Logan[2,5]*

[1]Department of Biochemistry and Biophysics, Stockholm University, Stockholm, Sweden; [2]Section for Biochemistry and Structural Biology, Centre for Molecular Protein Science, Department of Chemistry, Lund University, Lund, Sweden; [3]Department of Chemistry - Ångström Laboratory, Uppsala University, Uppsala, Sweden; [4]Structural Proteomics, SciLifeLab, Lund University, Lund, Sweden; [5]Cryo-EM for Life Science, SciLifeLab, Lund University, Lund, Sweden

**\*For correspondence:**
Derek.Logan@biochemistry.lu.se

†Deceased

**Competing interest:** The authors declare that no competing interests exist.

**Abstract** A small, nucleotide-binding domain, the ATP-cone, is found at the N-terminus of most ribonucleotide reductase (RNR) catalytic subunits. By binding adenosine triphosphate (ATP) or deoxyadenosine triphosphate (dATP) it regulates the enzyme activity of all classes of RNR. Functional and structural work on aerobic RNRs has revealed a plethora of ways in which dATP inhibits activity by inducing oligomerisation and preventing a productive radical transfer from one subunit to the active site in the other. Anaerobic RNRs, on the other hand, store a stable glycyl radical next to the active site and the basis for their dATP-dependent inhibition is completely unknown. We present biochemical, biophysical, and structural information on the effects of ATP and dATP binding to the anaerobic RNR from *Prevotella copri*. The enzyme exists in a dimer–tetramer equilibrium biased towards dimers when two ATP molecules are bound to the ATP-cone and tetramers when two dATP molecules are bound. In the presence of ATP, *P. copri* NrdD is active and has a fully ordered glycyl radical domain (GRD) in one monomer of the dimer. Binding of dATP to the ATP-cone results in loss of activity and increased dynamics of the GRD, such that it cannot be detected in the cryo-EM structures. The glycyl radical is formed even in the dATP-bound form, but the substrate does not bind. The structures implicate a complex network of interactions in activity regulation that involve the GRD more than 30 Å away from the dATP molecules, the allosteric substrate specificity site and a conserved but previously unseen flap over the active site. Taken together, the results suggest that dATP inhibition in anaerobic RNRs acts by increasing the flexibility of the flap and GRD, thereby preventing both substrate binding and radical mobilisation.

## eLife assessment

This study advances our understanding of the allosteric regulation of anaerobic ribonucleotide reductases (RNRs) by nucleotides, providing **valuable** new structural insight into class III RNRs containing ATP cones. The cryo-EM structural characterization of the system is **solid**, but some open questions remain about the interpretation of activity/binding assays and the HDX-MS results that have been newly incorporated compared to a previous version. The work will be of interest to

biochemists and structural biologists working on ribonucleotide reductases and other allosterically regulated enzymes.

## Introduction

Ribonucleotide reductases (RNRs) are a family of enzymes with sophisticated radical chemistries and allosteric regulation. RNRs produce all four deoxyribonucleotides and are the only enzymes providing de novo building blocks for DNA replication and repair in all free-living organisms (*Mathews, 2016*). Virtually all RNRs possess an allosteric site regulating substrate specificity (the s-site), a crucial aspect of RNR that ultimately provides a balanced supply of deoxyribonucleoside triphosphates (dNTPs) (*Mathews, 2016*; *Mathews, 2018*). A second allosteric site (the a-site) is located in an N-terminal domain called the ATP-cone (*Aravind et al., 2000*), found in the majority of RNRs (*Jonna et al., 2015*). Whereas the s-site binds all dNTPs and usually ATP, the a-site in the ATP-cone can only bind ATP and dATP. Binding of ATP activates RNR, and binding of dATP inhibits it (*Hofer et al., 2012*; *Martínez-Carranza et al., 2020*).

The RNR family consists of three evolutionarily related classes (I–III) with a common radical-based reaction mechanism but differing in the mode of radical generation and in quaternary structure (*Wiley, 2001*; *Lundin et al., 2015*). Class I, strictly aerobic RNRs are heterotetramers consisting of two α subunits (NrdA) with binding sites for substrates and allosteric nucleotides and two β subunits (NrdB) that harbour a radical initiator in the form of a stable radical or an oxidised metal cluster (*Wiley, 2001*). Class II RNRs are either dimers or monomers with an α subunit (NrdJ) that, in addition to binding sites for substrate and allosteric nucleotides, also harbours the radical initiator cofactor adenosylcobalamin. Class III RNRs, which only function anaerobically, are dimers of an α subunit (NrdD) with a stable radical close to the active site, and binding sites for allosteric nucleotides. The radical is introduced via encounter with a specific radical-SAM enzyme called NrdG, and once activated NrdD can perform multiple turnovers in the absence of NrdG (*Backman et al., 2017*; *Torrents et al., 2001*).

The dATP-dependent inhibition has been biochemically characterised in representatives of all three RNR classes, but only structurally studied in class I. A common denominator of all inhibited class I RNRs is oligomerisation, leading to disturbed radical transfer between the α and β subunits. Hitherto four different inhibitory oligomerisation mechanisms have been identified: heterooctameric α₄β₄ complexes in *Escherichia coli*, *Clostridium botulinum*, and *Neisseria gonorrhoeae* (*Ando et al., 2011*; *Martínez-Carranza et al., 2020*; *Wei et al., 2014*), an α₄ complex in *Pseudomonas aeruginosa* (*Johansson et al., 2016*), α₆ complexes in the eukaryotes *Homo sapiens*, *Saccharomyces cerevisiae*, and *Dictyostelium discoideum* (*Ando et al., 2016*; *Brignole et al., 2018*; *Crona et al., 2013*; *Fairman et al., 2011*), and a β₄ complex in the bacterium *Leeuwenhoekiella blandensis* (*Rozman Grinberg et al., 2018a*). All of these class I oligomers involve protein–protein interactions mediated by the ATP-cone domain.

The ATP-cone is in essence restricted to RNRs and the RNR-specific repressor NrdR (*Rozman Grinberg et al., 2022*). Among RNRs it is found in about 80% of class III enzymes and about 50% of class I enzymes, but less than 10% of class II enzymes (*Jonna et al., 2015*). The presence and function of the ATP-cone domain distinguish anaerobic RNRs from the other members of the large glycyl radical enzyme (GRE) family that are otherwise structurally related (*Backman et al., 2017*). The allosteric regulation mechanisms of class III RNRs (NrdDs) from *E. coli*, bacteriophage T4 and *Lactococcus lactis* were determined several decades ago (*Andersson et al., 2000*; *Eliasson et al., 1994*; *Torrents et al., 2000*). T4NrdD was the first class III RNR structure solved (*Logan et al., 1999*), and later, structures of *Thermotoga maritima* NrdD (TmNrdD) have been published (*Aurelius et al., 2015*; *Wei et al., 2014*). However, these two NrdDs lack an ATP-cone and the structural basis for allosteric activity regulation in the class III RNRs is still an outstanding question. Here, we present structural, biochemical, and biophysical studies on the ATP-cone-containing NrdD from the human pathogen *Prevotella copri* (PcNrdD; *Nii et al., 2023*), showing that the binding of dATP causes increased flexibility of the C-terminal glycyl radical-bearing domain, as well as a flap-like loop that binds across the top of the active site in the active form. This increase in dynamics is coupled to enzymatic inactivation by inhibition of substrate binding and radical transfer. The final outcome of dATP inhibition is thus blocked radical transfer in both class I and III RNRs, but it is achieved by completely different mechanisms, i.e.

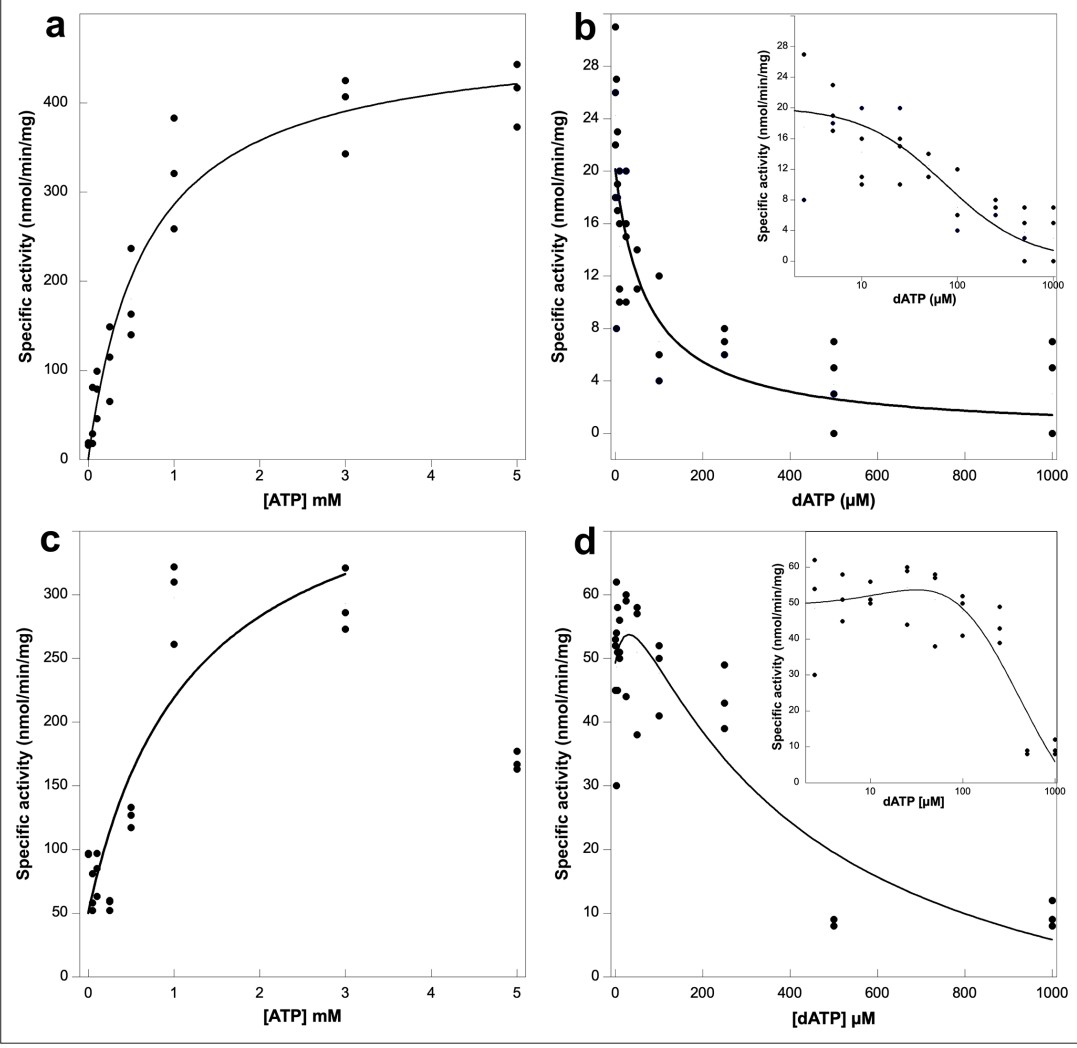

**Figure 1.** Activity assays of *Prevotella copri* NrdD in the presence of ATP or dATP. A 2.5-fold excess of holoNrdG over apoNrdD was used to study the allosteric regulation effect of ATP and dATP on the a-site. GTP reduction was monitored with 1 mM dTTP as effector in the s-site and titrated with ATP (**a**) or dATP (**b**) in the a-site. Cytidine triphosphate (CTP) reduction assays were titrated with ATP (**c**) or dATP (**d**), in this case acting both as s-site effectors and a-site regulators. Experiments were performed in triplicate for (**a**), (**c**), and (**d**) and with four replicates for (**b**). Insets in panels (**b**) and (**d**) show the results plotted in log scale. Curve fits for calculation of $K_L$ and $K_i$ used *Equations 1 and 2*, respectively, given in Materials and methods, and in (**d**) *Equation 3*. Curve fits for (**c**) and (**d**) used a start activity of 50 nmol/min/mg, and in (**c**) only results for 0–3 mM ATP were used. The $R$ values for curve fits in panels (**a–d**) were 0.99, 0.95, 0.83, and 0.96, respectively.

The online version of this article includes the following figure supplement(s) for figure 1:

**Figure supplement 1.** Formate requirement in PcNrdD.

**Figure supplement 2.** Reconstitution of the [Fe–S] cluster of the *Prevotella copri* NrdG.

oligomerisation prevents radical transfer between subunits in class I RNRs but long-range modulation of the flexibility of key structural elements prevents substrate binding in class III RNRs.

## Results

### Allosteric activity regulation by ATP and dATP

The amino acid sequence of PcNrdD suggests that it belongs to the formate-requiring class III RNRs (*Burnim et al., 2022b*; *Wei et al., 2014*), and initial optimisation of the assay composition showed that this was the case (*Figure 1—figure supplement 1*). ATP-cone-mediated activation of PcNrdD

enzyme activity by ATP or inhibition by dATP was tested with two different substrates. With guanosine triphosphate (GTP) as substrate the s-site was filled with dTTP as effector, and in absence of any a-site effector the basal level of GTP reduction was 21 ± 2 nmol/min/mg. An increasing concentration of ATP stimulated activity to a $k_{cat}$ of 1.3 s$^{-1}$ (478 ± 26 nmol/min/mg), with a $K_L$ of 0.67 ± 0.12 mM (*Figure 1a*). On the other hand, an increasing concentration of dATP resulted in an abrupt inhibition of enzyme activity with a $K_i$ of 74 ± 24 μM (*Figure 1b*). Hence, the $K_i$ value for dATP inhibition via the ATP-cone is approximately 10-fold lower compared to the $K_L$ value for ATP activation via the ATP-cone.

The activation/inhibition experiments were also carried out with CTP as substrate. Specificity effectors for CTP reduction are ATP or dATP, so in these experiments both the s- and a-sites were conceivably filled with ATP or dATP, respectively. It is obvious from *Figure 1c* that increasing concentrations of ATP initially stimulate activity and that high levels of ATP cause inhibition, plausibly by ATP also acting as substrate. We therefore used only results up to 3 mM ATP to calculate an approximate $k_{cat}$ of 1.1 s$^{-1}$ (385 ± 116 nmol/min/mg) with an apparent $K_L$ of 0.67 ± 0.52 mM (*Figure 1c*). Addition of dATP initially results in a stimulation of enzyme activity, conceivably when the s-site is filled with dATP, after which competing inhibition appears when dATP also binds to the ATP-cone (*Figure 1d*). All in all, these results suggest that the ATP-cone in PcNrdD responds similarly to inhibition by dATP as has been observed before for several class I RNRs.

## Binding of nucleotides to the ATP-cone

To confirm binding of ATP or dATP to the ATP-cone we used isothermal titration calorimetry (ITC) and microscale thermophoresis (MST) (*Figure 2*). Substrate GTP and s-site effector dTTP were used to fill up the other nucleotide-binding sites in PcNrdD. Binding of dATP occurred with a $K_D$ of 6 μM, whereas binding of ATP was fourfold weaker with a $K_D$ of 26 μM (*Figure 2a–c*). The number of bound dATP molecules observed with ITC was almost twofold higher for dATP compared to ATP, but both values were below 1. Nucleotide binding measured with MST confirmed our results for ATP (*Figure 2d*) that had a $K_D$ of 25 μM, and dATP that with this method had a $K_D$ of 23 μM (*Figure 2e*). However, in cryo-EM experiments, where higher nucleotide concentrations can be used, we show that the ATP-cone can bind two ATP or two dATP molecules (see below), as also expected based on its sequence. The $K_L$ and $K_i$ values reported above for ATP activation and dATP inhibition of GTP reduction conceivably reflect the requirement for two bound nucleotide molecules to achieve these effects, whereas the $K_D$ measurements via ITC and MST are restricted by the component concentrations that can be used and plausibly only reflect the first nucleotide bound.

It is noteworthy that the $K_D$ values for ATP and dATP do not differ by orders of magnitude, as has been found for several class I RNRs (*Ando et al., 2016*; *Birgander et al., 2004*; *Ormö and Sjöberg, 1990*; *Rozman Grinberg et al., 2018a*; *Rozman Grinberg et al., 2018b*; *Torrents et al., 2006*). Collectively, ITC and MST instead suggested that ATP and dATP may interact at two distinct sites in the ATP-cone for the functional readout of dATP inhibition, as has earlier been observed in the transcriptional repressor NrdR (*Rozman Grinberg et al., 2022*). To test this hypothesis, apo-PcNrdD was incubated with ATP only, dATP only, or a combination of ATP and dATP, then desalted, boiled, centrifuged, and the nucleotide content of the supernatant analysed by high performance liquid chromatography (HPLC). *Table 1* shows that only dATP was bound when PcNrdD was incubated with a combination of ATP and dATP. This was also the case when incubations were performed in the presence of s-site effector and substrate, and also when the desalting was performed in the presence of s-site allosteric effector and substrate. In incubations with only ATP, s-site effector and substrate were needed during the entire work-up procedure for ATP to be retained by the protein. In conclusion, the binding experiments suggest that the PcNrdD ATP-cone can bind either ATP or dATP but not both simultaneously. Importantly, the results point to allosteric communication between the s-site, the active site, and the a-site in the ATP-cone, as dTTP and GTP are required for any ATP to be retained. Both of these observations are consistent with the structural analyses shown below.

## Modulation of the oligomeric state by ATP and dATP in the presence of allosteric effectors

Binding of dATP to the ATP-cone in all class I RNRs studied to date results in formation of higher oligomers unable to perform catalysis (reviewed in *Martínez-Carranza et al., 2020*). We therefore next asked whether a similar mechanism would be valid for dATP-mediated inhibition of a class III RNR. To

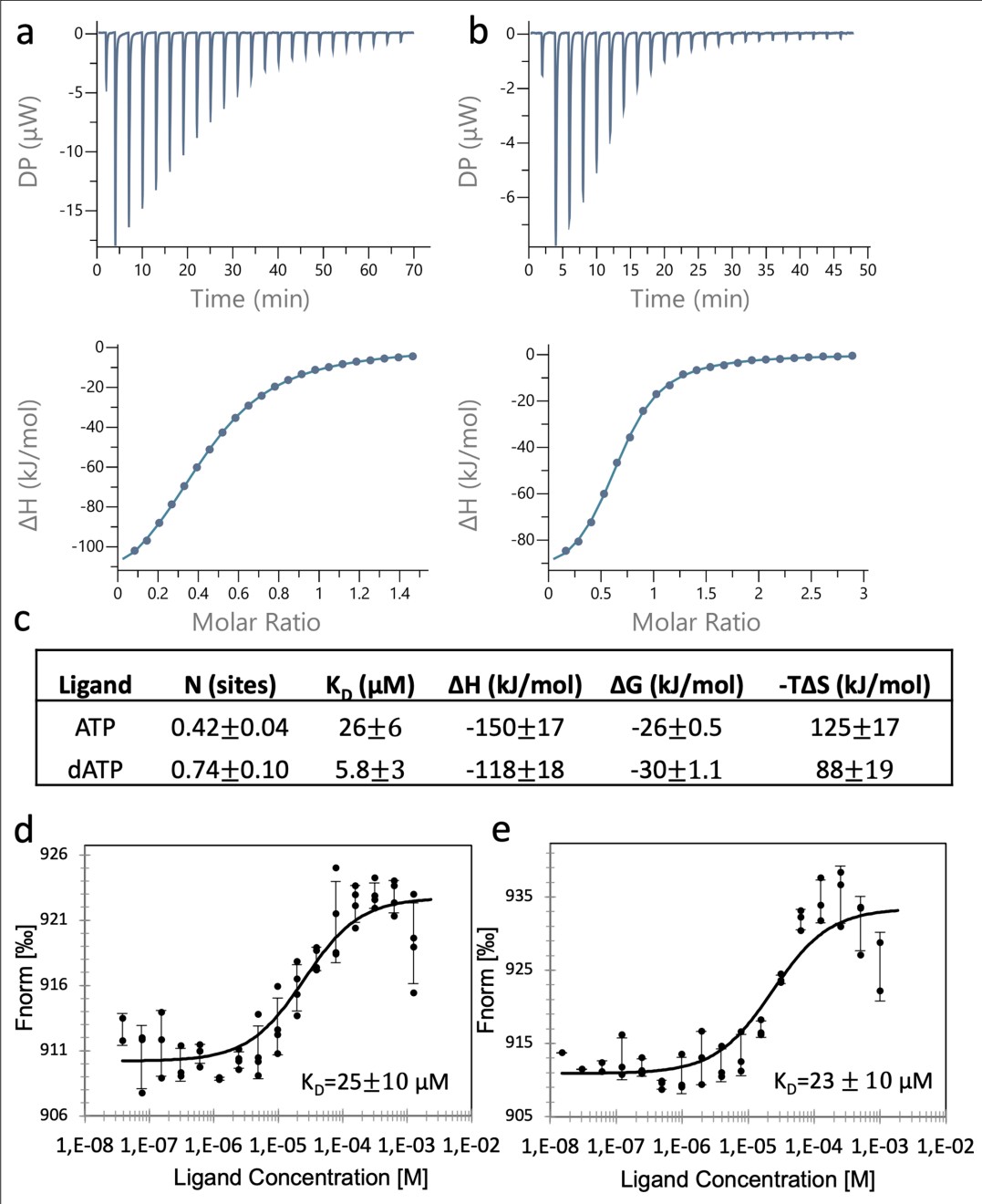

**Figure 2.** Binding of nucleotides to the ATP-cone. Binding assay using isothermal titration calorimetry (ITC): Representative thermograms of binding of ATP to 50 µM PcNrdD (**a**); and binding of dATP to 200 µM PcNrdD (**b**); (**c**) thermodynamic parameters of ligand binding. All titrations were performed in the presence of 1 mM GTP substrate and 1 mM s-site effector dTTP at 20°C. Binding of nucleotides to the ATP-cone using microscale thermophoresis (MST): (**d**) Binding of ATP and (**e**) binding of dATP. All MST-binding experiments were performed in the presence of 1 mM s-site effector dTTP and 5 mM GTP substrate at room temperature. Control experiments were done for ITC, by titrating the nucleotide into the buffer (not shown).

elucidate the oligomeric states of PcNrdD we used gas-phase electrophoretic mobility molecular analysis (GEMMA) (*Kaufman et al., 1996*), a method similar to electrospray mass spectrometry. It requires low protein and nucleotide concentrations as well as volatile buffers.

When loaded with ATP, PcNrdD was in a dimer–tetramer equilibrium shifted towards dimers (*Figure 3a*; *Figure 3—figure supplement 1*; *Table 2*). Addition of the CTP substrate or the s-site

**Table 1.** Amount of ATP and/or dATP bound to PcNrdD under different incubation and desalting conditions.

| Incubating conditions | Desalting conditions | ATP (mol/mol NrdD) | dATP (mol/mol NrdD) |
|---|---|---|---|
| NrdD + ATP* | | 0.09*** | 0 |
| NrdD + dATP[†] | No nucleotides during desalting | 0 | 0.97[†††] |
| NrdD + ATP + dATP [‡] | | 0.01 | 0.89 [‡‡‡] |
| NrdD + ATP + dTTP + GTP [§] | | 0.03 | 0.01 |
| NrdD + dATP + dTTP + GTP [¶] | No nucleotides during desalting | 0 | 0.64 |
| NrdD + ATP + dATP + dTTP + GTP** | | 0.01 | 1 |
| NrdD + ATP + dTTP + GTP[††] | | 0.35 | 0 |
| NrdD + dATP + dTTP + GTP [‡‡] | dTTP + GTP included during desalting [¶¶] | 0 | 0.52 |
| NrdD + ATP + dATP + dTTP + GTP [§§] | | 0.05 | 0.85 |

*1 mM ATP.

[†]1 mM dATP.

[‡]1 mM ATP, 1 mM dATP.

[§]3 mM ATP, 1 mM dTTP, 5 mM GTP.

[¶]1 mM dATP, 1 mM dTTP, 5 mM GTP.

**3 mM ATP, 1 mM dATP, 1 mM dTTP, 5 mM GTP.

[††]3 mM ATP, 2 mM dTTP, 5 mM GTP.

[‡‡]1 mM dATP, 2 mM dTTP, 5 mM GTP.

[§§]3 mM ATP, 1 mM dATP, 2 mM dTTP, 5 mM GTP.

[¶¶]0.1 mM dTTP, 1 mM GTP.

***Summary of 0.07 mol/mol NrdD ATP and 0.02 mol/mol NrdD ADP.

[†††]Summary of 0.78 mol/mol NrdD dATP, 0.18 mol/mol NrdD dADP, and 0.01 mol/ molNrdD dAMP.

[‡‡‡]Summary of 0.70 mol/mol NrdD dATP, 0.18 mol/mol NrdD dADP, and 0.01 mol/mol NrdD dAMP.

effector dTTP resulted in a similar equilibrium. In contrast, the dimer–tetramer equilibrium of dATP-loaded PcNrdD was shifted towards tetramers, and this equilibrium was likewise not affected by addition of the CTP substrate (*Figure 3b*; *Figure 3—figure supplement 1*, *Table 2*). Hence, dATP inhibition of enzyme activity seems to work differently in PcNrdD than the clear-cut and drastic change in oligomeric structure observed for class I RNRs, since both ATP- and dATP-loaded forms of PcNrdD were in dimer–tetramer equilibria.

## Glycyl radical formation in presence of ATP or dATP

Next, we asked whether dATP inhibition was mediated by prevention of glycyl radical formation. Apo-PcNrdD was incubated with NrdG and *S*-adenosylmethionine in presence of ATP or dATP. Whereas reduction of substrate requires the presence of formate, the formation of a glycyl radical does not. We therefore incubated with or without formate as well as with or without substrate. *Figure 4* shows that the glycyl radical was readily formed in the presence of dATP both with and without the CTP substrate, and the result was similar in the absence of formate (*Figure 4—figure supplement 1*). Interestingly, under all conditions the glycyl radical content was higher in the presence of dATP compared to with ATP, and it was also more stable than in the absence of any effector nucleotides. In addition, the shape of the doublet electron paramagnetic resonance (EPR) signal was similar under all conditions studied. Collectively, these results show that the glycyl radical is readily formed in the presence of a dATP concentration that inhibits RNR activity.

## Substrate binding to PcNrdD

To study substrate binding to activated and inhibited PcNrdD we used MST with labelled protein and GTP or CTP as substrate. The results were clear-cut in both sets of experiments (*Figure 5*). ATP-loaded

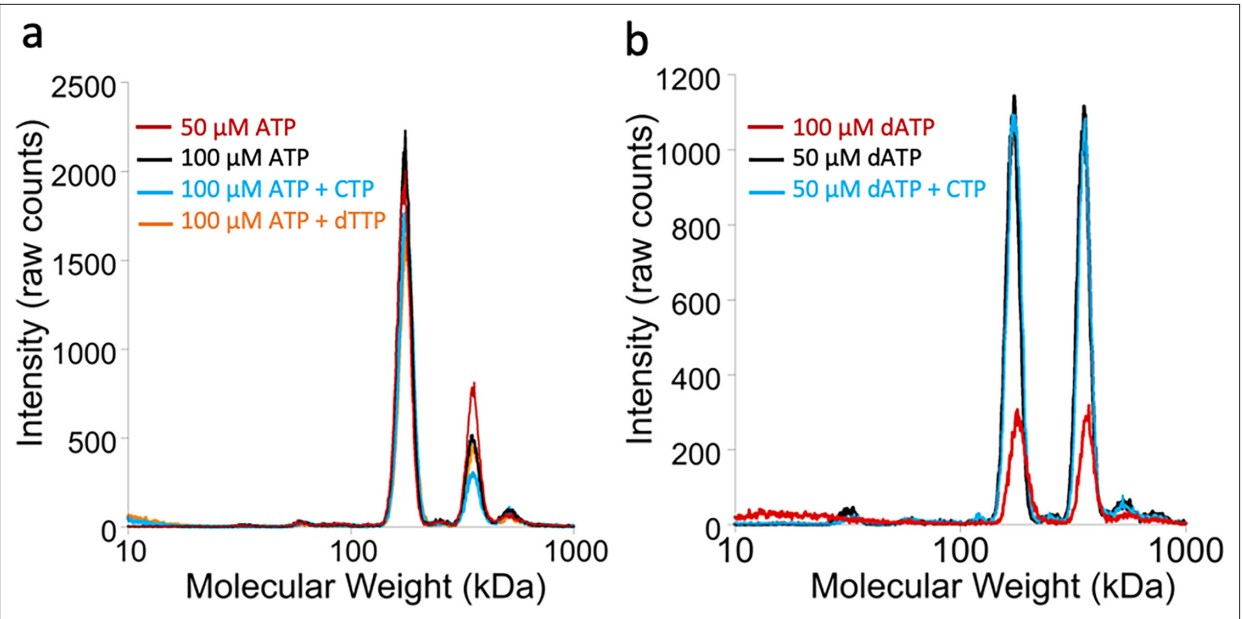

**Figure 3.** Oligomeric states of *Prevotella copri* NrdD in the presence of nucleotides determined by gas-phase electrophoretic mobility molecular analysis (GEMMA). (**a**) Apo-PcNrdD (2 µM) loaded with the activator ATP (50–100 µM) (black and red) in the presence of CTP (100 µM) as a substrate (cyan) or dTTP (100 µM; orange) as the allosteric effector. (**b**) apo-PcNrdD (2 µM) loaded with the inhibitor dATP (50–100 µM) (black and red) in the presence of CTP (50 µM) (cyan) as the substrate. Each sample was scanned five times to increase signal-to-noise level. (**c**) Calculated numbers of monomers based on measured molecular weight and fractions of dimers versus tetramers after conversion of experiments (**a**) and (**b**) to mass concentrations.

The online version of this article includes the following figure supplement(s) for figure 3:

**Figure supplement 1.** Size exclusion chromatography (SEC) analyses of *Prevotella copri* NrdD in the presence of nucleotides.

PcNrdD bound GTP in the presence of s-site effector dTTP (*Figure 5a*), and it bound CTP in the presence of only ATP (*Figure 5b*), whereas dATP-loaded PcNrdD had extremely low affinity to GTP and CTP under similar conditions (*Figure 5c, d*). These results show that the dATP inhibition of enzyme activity in a class III RNR is mediated by inhibited binding of substrate.

## Cryo-EM structure of the ATP–CTP-bound PcNrdD dimer

Two-dimensional (2D) classes of particles extracted from micrographs of PcNrdD incubated with effector ATP and substrate CTP indicated only dimers, with no appreciable amounts of tetramer. From a set of 570,730 particles we obtained a map from cryoSPARC at 3.0 Å. Application of deep learning local sharpening in DeepEMhancer (*Sanchez-Garcia et al., 2021*) improved the quality of the map, and we were able to build a reliable model for the great majority of the structure. As the ATP-cone density still did not permit reliable modelling, three-dimensional (3D) classification was carried out

**Table 2.** Calculated numbers of monomers based on measured molecular weight and fractions of dimers versus tetramers after conversion of experiments (a) and (b) to mass concentrations.

| Effector(s) | Mw (kDa) | No. of monomers | Dimers vs. tetramers (%) |
|---|---|---|---|
| ATP (50 µM) | 176/358 | 2.1/4.2 | 50/50 |
| ATP (100 µM) | 178/360 | 2.1/4.3 | 60/40 |
| ATP (100 µM)+CTP | 179/356 | 2.1/4.3 | 64/36 |
| ATP (100 µM)+dTTP | 171/348 | 2.1/4.2 | 60/40 |
| dATP (50 µM) | 167/363 | 2.0/4.3 | 32/68 |
| dATP (100 µM) | 179/361 | 2.1/4.3 | 32/68 |
| dATP (50 µM)+CTP | 175/358 | 2.1/4.2 | 34/66 |

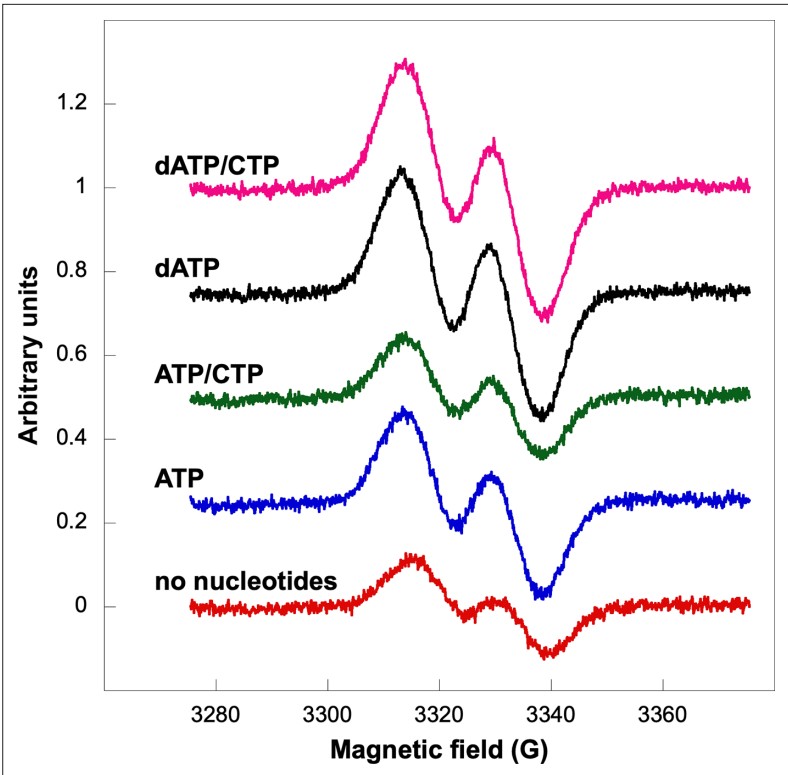

**Figure 4.** Glycyl radical formation after 20 min in presence of formate and ATP ± CTP or dATP ± CTP. Nucleotide concentrations were: 1.5 mM ATP, 1 mM dATP, and 1 mM CTP. Traces are arbitrarily moved to increase visibility and scaled to identical units (Y-axes).

The online version of this article includes the following figure supplement(s) for figure 4:

**Figure supplement 1.** Glycyl radical formation in absence of formate after 20-min incubation.

on this particle set, enabling the identification of a subset of 291,231 particles that gave a volume at 3.2 Å with better-defined ATP-cones (*Figure 6a*). This map was used for refinement of the final model (*Figure 6—figure supplement 1*).

PcNrdD is a compact dimer (*Figure 6b*). Each monomer consists of 739 residues, of which residues 1–91 constitute the ATP-cone, 92–110 a linker region, 111–122 a flap that folds over the substrate in the active site, 123–671 are the core domain with its 10-stranded α/β barrel fold typical of the RNR/GRE family, and 676–739 are the C-terminal glycyl radical domain (GRD) that contains a structural $Zn^{2+}$ site followed by the buried loop containing the radical glycine residue at position Gly711 and a C-terminal tail that extends across the surface of the protein (*Figure 6b, c*). The closest structural neighbours as identified by the DALI server (*Holm, 2022*) are the NrdDs from bacteriophage T4 (*Logan et al., 1999*) (1H7A, 2.5 Å root mean square deviation [rmsd] in Cα positions for 535 residues, 27% sequence identity) and *T. maritima* (*Aurelius et al., 2015*; *Wei et al., 2014*) (4U3E, 3.0 Å rmsd for 508 residues, 16% sequence identity; *Table 3*). The linker, flap, and GRD are well-ordered in one of the monomers (dark blue in *Figure 6*) and substrate CTP is bound (*Figure 6e*), while in the other monomer this region is less well-ordered and the active site is empty (*Figure 6—figure supplements 2 and 3*). We will refer to these as the 'active' and 'inactive' monomers, respectively.

At the concentration of ATP used (1 mM), ATP is not observed to bind to the s-sites near the dimer interface. The ATP-cones containing the a-site are flexible, but the reconstruction from a subset of particles allowed the modelling of two ATP molecules per ATP-cone and many of the most important side chains. The ATP molecules are bound similarly to the twin dATP molecules seen in the ATP-cones of *P. aeruginosa* NrdA (PaNrdA) and *L. blandensis* NrdB (LbNrdB) (*Johansson et al., 2016*; *Rozman Grinberg et al., 2018a*), and to dATP bound to PcNrdD itself (see below). However, the ATP-cones in PcNrdD exhibit a hitherto unobserved relative orientation in which the four triphosphate groups are projected towards each other (*Figure 6b, d*). The large negative charge is compensated for by the

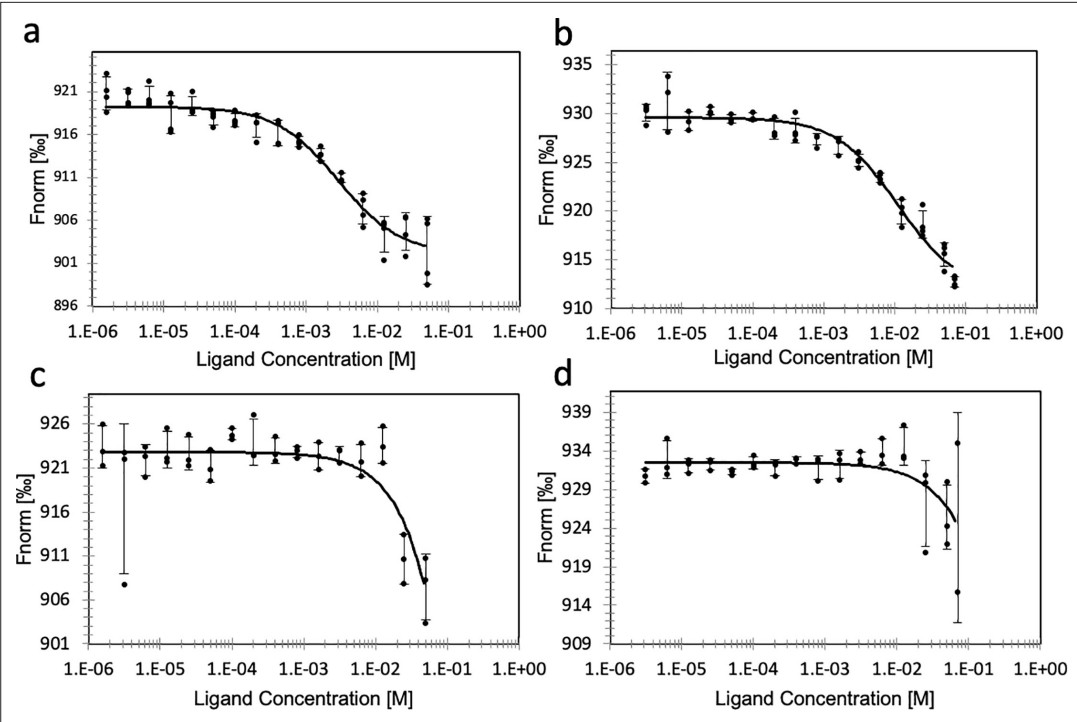

**Figure 5.** Binding of substrate to ATP- or dATP-loaded PcNrdD. Binding of GTP (**a**) and binding of CTP (**b**) to ATP-loaded PcNrdD. (**c**) Binding of GTP and (**d**) binding of CTP to dATP-loaded PcNrdD. No additional nucleotides were present in CTP-binding experiments, whereas binding of GTP was performed in the presence of 1 mM s-site effector dTTP. Fitted $K_D$s are 2.8 ± 0.5 and 11 ± 2.4 mM for GTP and CTP binding, respectively, in the presence of ATP. In the presence of dATP fitted $K_D$s were ≥2833 and ≥803 mM for GTP and CTP binding, respectively. Experiments were performed at room temperature.

proximity of as many as 12 Arg and Lys side chains from both monomers. The ATP-cones interact by contacts involving the 'roof' of the domain (residues 1–18). Arg11 of each ATP-cone reaches over to interact with the α-phosphate group of one of the ATP molecules in the other ATP-cone, and a salt bridge is formed between Asp9 at the end of the lid and Arg91 at the end of the last helix of the other ATP-cone. Strikingly, the dimer of ATP-cones is slightly offset from the dimer axis of the core protein, such that the ATP-cone of the inactive monomer approaches the core domain of the active monomer, while the ATP-cone of the active monomer is more distant from the core domain of the inactive one (**Figure 6a, b**). However, the closest atom of any ATP molecule is over 30 Å from the substrate CTP in the active site.

The glycyl radical site is found at Gly711, at the tip of a loop projected from the C-terminal region of the enzyme into the barrel, where it approaches the radical initiator cysteine Cys416 at the tip of a loop stretching through the barrel from the other side (**Figure 7**). The second cysteine necessary for the radical mechanism of PcNrdD is Cys171 on the first β-strand of the barrel (**Figure 6e**). Consistent with the fact that PcNrdD belongs to the group of anaerobic RNRs that use formate as overall reductant (**Burnim et al., 2022a**; **Mulliez et al., 1995**; **Wei et al., 2014**), there is no third cysteine residue on the sixth β-strand, and its place is taken by Asn435. As usual for GREs, Gly711 is completely buried within the barrel, with no solvent-accessible surface area. The base, phosphate, and most of the ribose of substrate CTP are well-defined but the density is weaker around the 5' C-atom (**Figure 6e**).

After the glycyl radical loop emerges from the α/β barrel, it traverses the top of the barrel, forming a short helix from residues 723 to 730 and ending in an extended tail. This conformation of the C-terminus is very similar to the one seen for TmNrdD in PDB entry 4U3E (**Wei et al., 2014**). This tail makes few specific interactions with other residues in the core.

When compared to previously determined NrdD structures, the first common structural element is a long α-helix in the core domain stretching from Thr123 to Leu141. Between this and the ATP-cone are the linker (92–110), followed by the flap over the substrate in the active site (111–122). Two

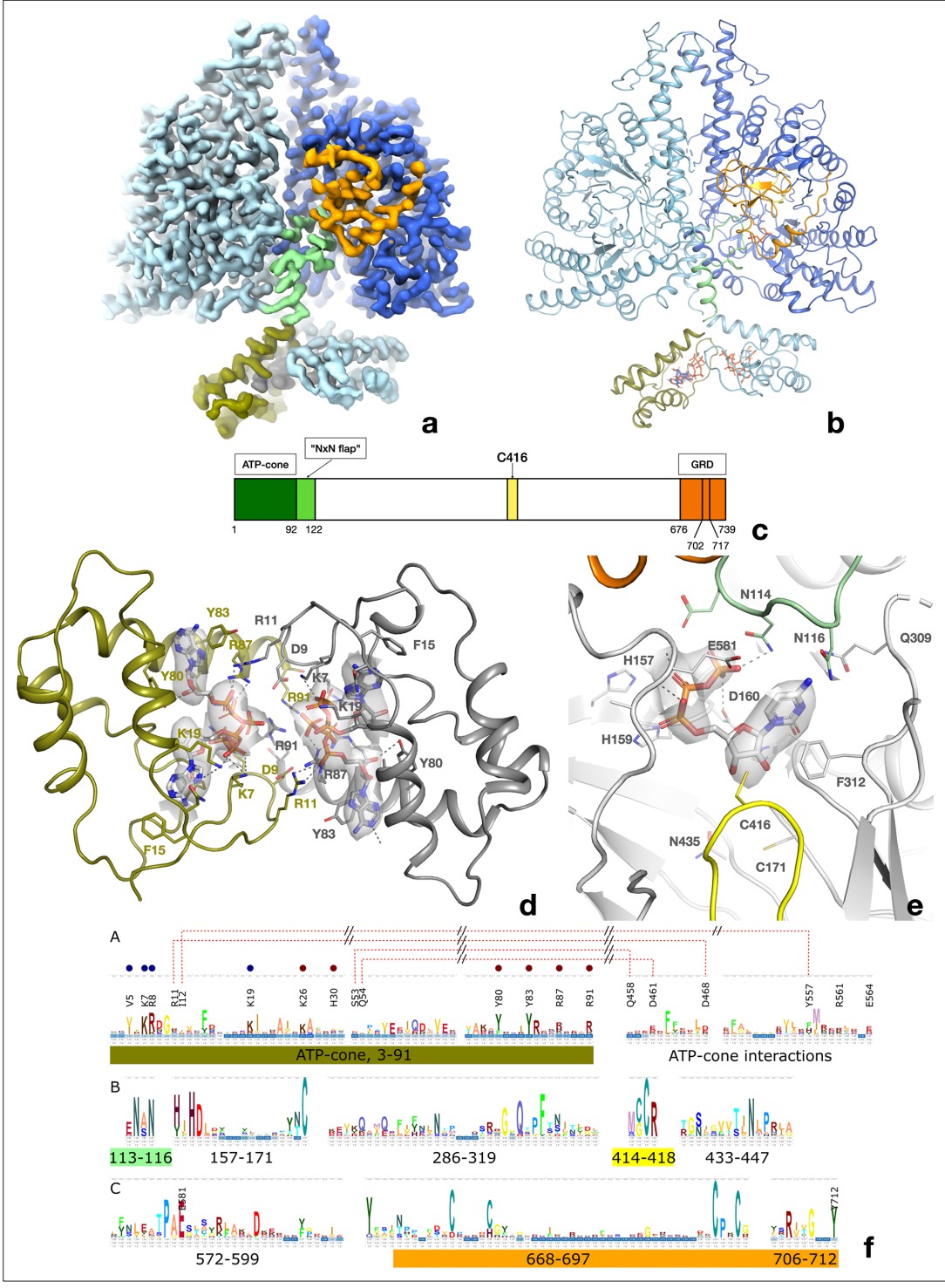

**Figure 6.** Structure of the PcNrdD dimer in complex with effector ATP and substrate CTP. (**a**) Cryo-EM map with C1 symmetry showing the best-ordered ATP-cones after post-processing using DeepEMhancer. For the right-hand (active) monomer, the ATP-cone (residues 1–91) and the linking helix (92–104) are coloured olive, the linker and NxN flap region pale green, the glycyl radical loop red, and the C-terminal extended region orange. The loop in the middle of the α/β barrel containing the radical initiator cysteine Cys416 at its tip is yellow. (**b**) Overview of the PcNrdD dimer with ATP and CTP. The two monomers of the dimer are coloured in different shades of grey. The ATP and CTP molecules are shown as sticks. (**c**) Schematic of the domain organisation of PcNrdD with the same colour scheme as (**a**) and (**b**); (**d**) Closeup view of the binding of four ATP molecules to the dimer of ATP-cones in PcNrdD. The view is from the bottom of the molecule as seen in (**a**) and (**b**). Disordered loops are shown as dotted

*Figure 6 continued on next page*

*Figure 6 continued*

lines. (**e**) Closeup view of the active site including the cryo-EM map for CTP. Residues within 4 Å of CTP are shown as sticks and polar interactions as dotted lines. (**f**) Sequence logos of NrdD sequence motifs. (**A**) ATP-cone plus downstream interaction partners, (**B**) central parts of sequence and (**C**) C-terminal parts. The numbering is from PcNrdD. Segments were selected to illustrate amino acids discussed in the text.

The online version of this article includes the following figure supplement(s) for figure 6:

**Figure supplement 1.** Cryo-EM data processing workflow for PcNrdD in the presence of ATP–CTP.

**Figure supplement 2.** The PcNrdD–ATP–CTP structure coloured by *B*-factor.

**Figure supplement 3.** Cryo-EM maps for the empty and occupied active sites of PcNrdD.

(**a**) Cryo-EM map for one of the four active sites of the dATP–CTP tetramer. The refined model is superposed. All side chains are shown as sticks. The finger loop is coloured yellow. (**b**) Map for one of the two active sites of the dATP–CTP dimer. (**c**) Complete density for the occupied active site of the ATP–CTP complex for comparison. The NxN flap is coloured green and the C-terminal region orange.

residues from the flap make H-bonds to CTP: Asn114 to the γ-phosphate group and Asn116 to the amino group of the cytosine base. We call this region the 'NxN flap', as the NxN motif (114–116) is one of the most highly conserved sequence motifs in the NrdD family (*Figure 6f*). This is the first time such interactions have been observed in an NrdD, as the flap corresponds to a disordered segment of 17 residues in the structures of TmNrdD and T4NrdD. However, an AlphaFold2 (*Jumper et al., 2021*) model of TmNrdD (entry Q9WYL) suggests that this linker potentially has the same conformation in TmNrdD and possibly all other NrdDs, whether or not they have an ATP-cone. As seen previously in TmNrdD, the triphosphate moiety is recognised through the conserved HxHD motif (157–160), H-bonds to the main chain atoms of a loop (577–581) following the third-last strand of the α/β barrel and the dipole of the following helix (*Figure 6f*).

Significantly, the NxN flap forms the nexus of a network of interactions (*Figure 7*) between the substrate, the linker to the ATP-cone, the C-terminal GRD, and 'loop 2' (residues 304–311) that is responsible for communicating the substrate specificity signal from the s-site at the dimer interface to the active site (*Aurelius et al., 2015*; *Larsson et al., 2001*; *Logan et al., 1999*). Important interactions linking the NxN flap to the C-terminus include H-bonds from Tyr712, which immediately follows the radical Gly711, to Glu581, which bridges to both the substrate and Ala115 in the flap, as well as from Arg719 in the GRD to the main chain carbonyl groups of Asn114 and Met117 in the flap. Tyr712 is highly conserved in the NrdD family (*Figures 6f and 7*). This nexus is crucial for allosteric activity regulation, as will be discussed later.

As the NxN flap appears to close off the top of the active site, we used the web server Caver Web v1.1 (*Stourac et al., 2019*) to analyse substrate access to the active site in the ATP-bound form, and identified a tunnel ~12 Å long leading from the surface of the protein to the active site that is 5.6 Å in diameter at its narrowest point (*Figure 7—figure supplement 1*). Thus, in principle a fresh substrate could diffuse into the active site on each catalytic cycle.

## Cryo-EM structure of the ATP–dTTP–GTP dimer

In order to probe the generality of the conformations observed in the ATP–CTP complex, we also solved the structure of PcNrdD in complex with ATP in the a-site, specificity effector dTTP in the s-site, and substrate GTP in the active site (c-site). Again, 2D classes show a predominantly dimeric form with no more than ~10% tetramers. From a set of 437,886 particles selected from 2D classes with the best apparent density for the ATP-cone region, we obtained a volume at 2.40 Å resolution for the dimer with C1 symmetry that was further improved using DeepEMhancer (*Figure 8—figure*

**Table 3.** Structural similarity comparisons to PcNrdD.
Structural similarity tables from DALI (PDB90, i.e. targets filtered at 90% sequence identity).

| PDB ID | Protein | Z-score | Residues aligned | RMSD (Å) | Sequence identity (%) |
|---|---|---|---|---|---|
| 1H7A | Bacteriophage T4 NrdD | 34.6 | 535 | 2.5 | 27 |
| 4U3E | *T. maritima* NrdD | 25.5 | 508 | 3.0 | 16 |
| 4COL | *T. maritima* NrdD / dATP | 24.8 | 493 | 2.9 | 17 |

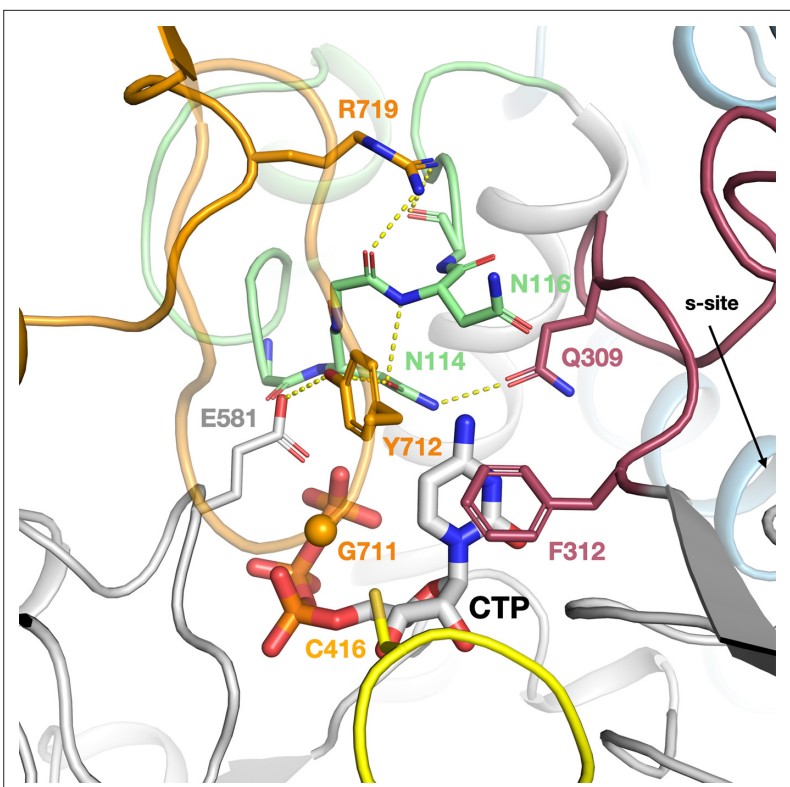

**Figure 7.** The intricate network of interactions between the NxN flap, C-terminal glycyl radical domain (GRD), loop 2, and substrate in the ATP–CTP complex of PcNrdD. The NxN flap is coloured light green, the GRD orange, loop 2 dark red, and the Cys radical loop yellow. The Gly radical loop is semi-transparent for clarity. The CA atom of Gly711 is indicated by an orange sphere. Important hydrogen bonds are shown as dotted lines.

The online version of this article includes the following figure supplement(s) for figure 7:

**Figure supplement 1.** The tunnel leading from the active site to the surface of PcNrdD in the ATP–CTP complex.

supplement 1). dTTP is clearly visible in the s-site (*Figure 8—figure supplement 2a*) as is GTP in the active site (*Figure 8—figure supplement 3a*). A $Mg^{2+}$ ion can be modelled in the s-site, coordinated by Glu290 and all three phosphate groups of dTTP. A β-hairpin loop between residues 185 and 192 that is disordered in the ATP–CTP structure in the absence of an s-site nucleotide is here ordered, due to an H-bond to the 2′-OH group of dTTP. Readout of the s-site nucleotide's identity is achieved through an H-bond to Asn298 (*Figure 8—figure supplement 2a*).

Like the ATP–CTP complex, the ATP–dTTP–GTP complex can also be partitioned into an active and an inactive monomer. The ATP-cone region is very flexible, but the last helix of the ATP-cone in the active monomer can be traced back to residue 85 (*Figure 8a, b*). The helix is not kinked at residue 91 as in the ATP–CTP dimer but forms an uninterrupted helix between residues 85 and 104. The volume for the ATP-cone region lies entirely on one side of the PcNrdD dimer axis, as is also apparent in some of the 2D classes (*Figure 8—figure supplement 4*). The ATP-cone cannot be modelled in detail, but if the volume is contoured at very low level, there is almost enough density for one ATP-cone. Taken together, this suggests that there is one partially disordered ATP-cone and one highly flexible in the ATP–dTTP–GTP complex. In the active monomer, the NxN flap and entire C-terminus are ordered while in the inactive monomer, the entire assembly of linker and NxN flap up to residue 120 and the entire GRD are so flexible that they cannot be seen in the cryo-EM volumes. We expect that the GRD behaves largely as a rigid body whose displacements relative to the core domain become larger in the dATP-bound form.

Intriguingly, binding of dTTP to the s-sites at the dimer interface induces two distinct conformations of loop 2 (*Figure 8c*). In the active monomer, loop 2 is projected towards the aforementioned network of interactions between NxN flap, GRD, and substrate, packing against the flap, though no polar interactions are apparent. However, loop 2 also makes extensive interactions with its equivalent

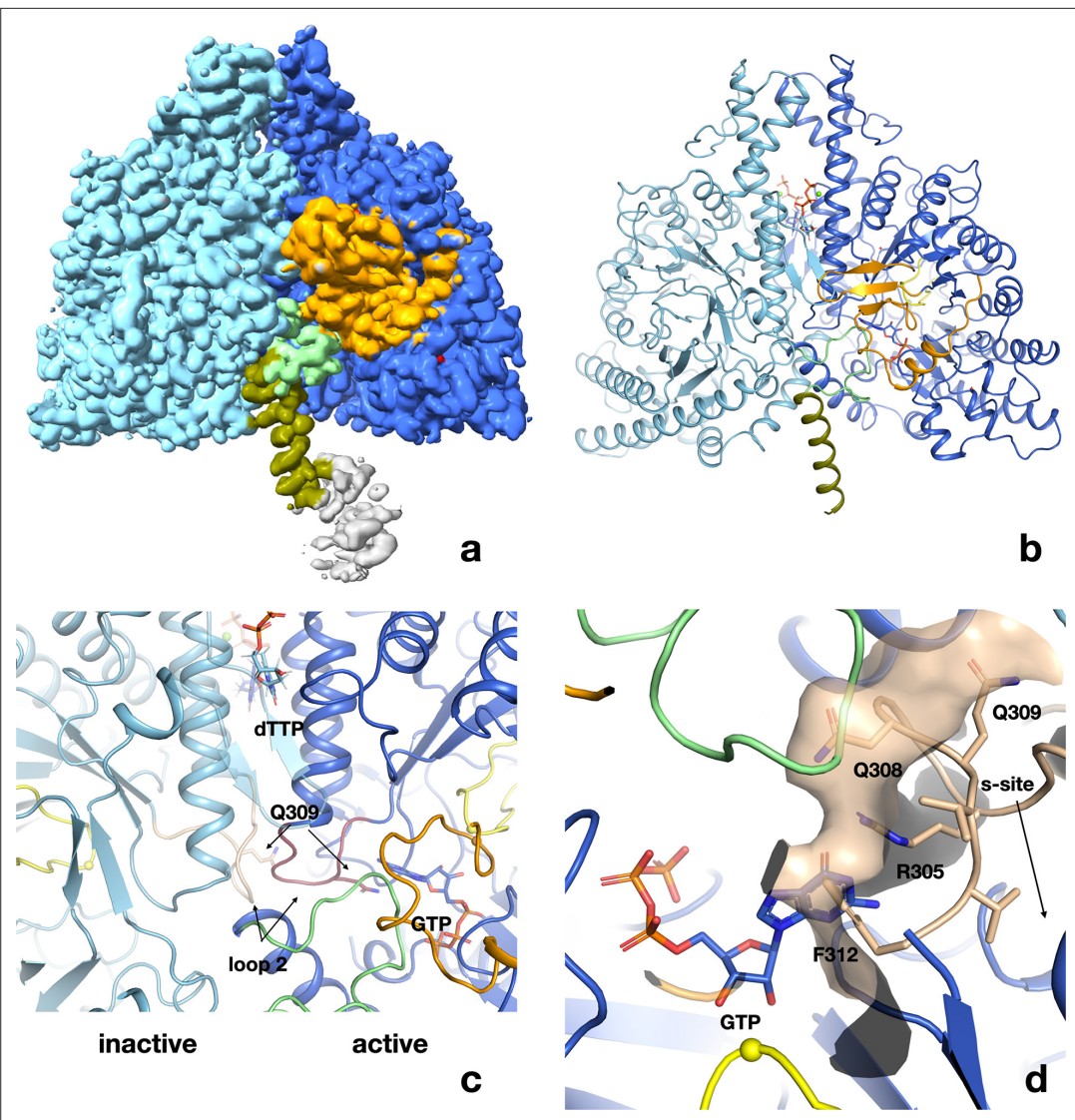

**Figure 8.** Structure of PcNrdD in complex with a-site effector ATP, s-site effector dTTP, and substrate GTP.
(**a**) Volume for the PcNrdD dimer in complex with ATP, dTTP, and GTP. The volume is contoured at a low level to emphasise the weak density for the more ordered ATP-cone domain in the active monomer. (**b**) Ribbon diagram of the ATP–dTTP–GTP complex coloured as in **Figure 6b**, except that the most N-terminal helix (olive) now extends from residues 85 to 104. (**c**) Zoom in on panel (**b**) to illustrate the highly asymmetrical loop 2 conformations in the active and inactive monomers. Loop 2 in the inactive monomer (light blue) is coloured wheat, while in the active monomer (dark blue) it is purple. The conformation of loop 2 that forms a complementary cradle for the guanosine base of the substrate GTP in the active monomer induces a conformation of the other loop 2 that precludes substrate binding. (**d**) Severe steric clash of loop 2 of the inactive monomer with the substrate and NxN flap. The view is rotated approximately 180° from panels a–c. Loop 2 from the inactive monomer (wheat) is superimposed on the active site of the active monomer (dark blue). The molecular surface of the loop is shown to emphasise that this conformation is incompatible with an ordered NxN flap (light green) and substrate binding. The Cys radical loop is yellow and Cys416 is marked by a sphere. The Gly radical loop is omitted for clarity.

The online version of this article includes the following figure supplement(s) for figure 8:

**Figure supplement 1.** Cryo-EM data processing workflow for PcNrdD in the presence of ATP–dTTP–GTP.

**Figure supplement 2.** Binding of allosteric substrate specificity nucleotides to the s-site at the dimer interface of PcNrdD.

**Figure supplement 3.** Cryo-EM map and interactions of (**a**) GTP in the active site of the PcNrdD–ATP–dTTP–GTP complex; (**b**) ATP in the PcNrdD–ATP–dGTP complex.

*Figure 8 continued on next page*

*Figure 8 continued*

**Figure supplement 4.** Representative two-dimensional (2D) classes showing side views of PcNrdD with diffuse density in the ATP-cone region.

**Figure supplement 5.** Cryo-EM data processing workflow for PcNrdD in the presence of ATP and dGTP.

**Figure supplement 6.** The highly asymmetrical loop 2 conformations in the active and inactive monomers of the PcNrdD–dGTP–ATP complex.

in the inactive monomer, which forces the latter into a conformation where it would sterically clash with an ordered NxN flap (*Figure 8d*). Furthermore, Phe412, which stacks on the substrate base in the active monomer, and Arg305, which is projected from the substrate-distal side of loop 2, sterically prevent the substrate binding in the inactive monomer. These may be important contributing factors to the disorder of the NxN flap and GRD. Finally, Gln309, which interacts with the substrate in the active monomer (*Figure 8c*), is oriented away from the substrate through its interactions with the 'active' loop 2.

## Cryo-EM structure of the ATP–dGTP dimer

As a third insight into the ATP-bound forms of PcNrdD, we solved the structure of PcNrdD in complex with specificity effector dGTP in the s-site and ATP at a concentration where it would both act as allosteric effector at the a-site and as the cognate substrate for dGTP (*Figure 8—figure supplement 5*). Density for dGTP is clearly visible in the s-site (*Figure 8—figure supplement 2b*) and for ATP in the active site (*Figure 8—figure supplement 3b*). Again, the two monomers can be divided into an active one and an inactive one. The conformations of loop 2 are once again asymmetrical (*Figure 8—figure supplement 6*). The conformation in the active monomer is well-defined, while loop 2 in the inactive monomer is more flexible than in the ATP–dTTP–GTP complex, but still compatible with the hypothesis that the asymmetric arrangement of loop 2 is responsible for disorder of the NxN–GRD–substrate network in the inactive monomer.

## Cryo-EM structure of the dATP-bound dimer

In micrographs of PcNrdD samples in the presence of 0.5 mM dATP, we observed an approximately 1:1 mixture of dimeric and tetrameric particles, consistent with results from GEMMA. To resolve whether dATP inhibition was linked to oligomerisation, we resolved the structures of each of these oligomers separately. We made a reconstruction of the dATP-bound PcNrdD dimer from 1,009,021 particles to 2.7 Å resolution that was further improved by DeepEMhancer postprocessing (*Figure 9—figure supplement 1*). The core domain is very similar to that of the ATP–CTP complex, with an rmsd in Cα positions of 1.2 Å for 1089 Cα atoms considering each dimer as a whole. dATP is clearly visible in the specificity site (*Figure 8—figure supplement 2*).

Strikingly, in contrast to the ATP complexes, the entire GRD is disordered in both monomers of the dATP-bound dimer (*Figure 9—figure supplement 2*). No structure is visible after residue 676 at the end of the last strand of the barrel. This suggests that inhibition by binding of dATP to the ATP-cone is coupled to disordering of the C-terminal domain. The NxN flap that covers the top of the substrate in the ATP/CTP-bound form is also disordered, as are the ATP-cones themselves.

## Cryo-EM structure of the dATP-bound tetramer

From the same micrographs that revealed the dATP-bound dimers we also resolved dATP-bound tetramers. We obtained a map from cryoSPARC at 2.8 Å, which was again improved with DeepEMhancer (*Figure 9a*; *Figure 9—figure supplement 1*). Two dATP molecules are bound to each ATP-cone and one at each of the specificity sites, giving a total of 12 visible dATPs per tetramer. The tetramer reveals an arrangement of two dimers (chains A/B and C/D, respectively) in which the ATP-cones are all well-ordered and one ATP-cone from each pair mediates interactions between the dimers (*Figure 9b*). The ATP-cones have an almost domain-swapped arrangement within each dimer relative to the core domains. This oligomeric arrangement has not previously been observed within the RNR family.

dATP binds to the specificity site as in the dimeric dATP-bound form (*Figure 8—figure supplement 2c*). The two dATP molecules bound to each ATP-cone do so very much as they do to the

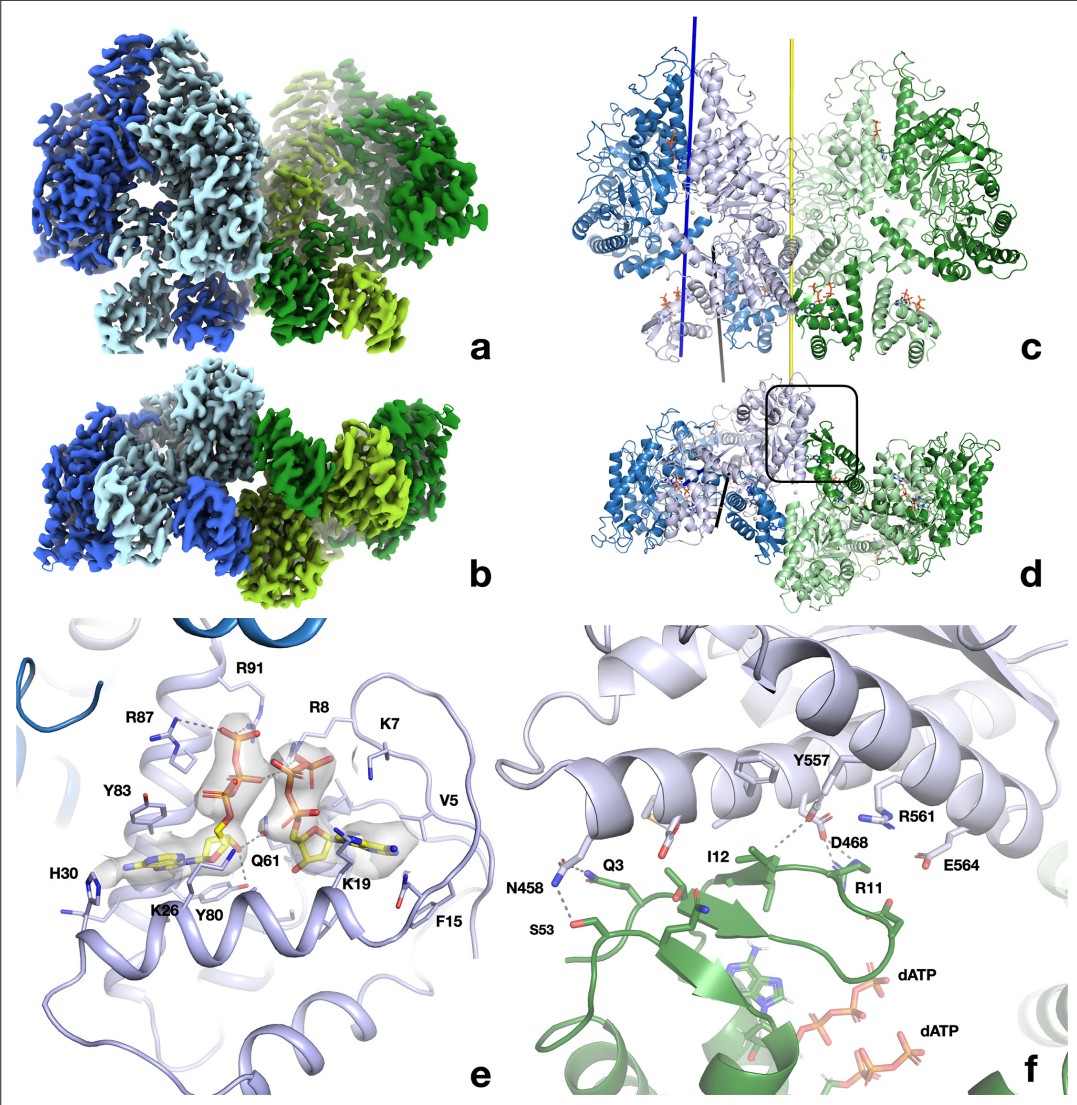

**Figure 9.** Structure of the dATP-bound PcNrdD tetramer. (**a**) Cryo-EM reconstruction coloured by chain: dimer 1 light blue and pale blue; dimer 2 light green and pale green. The four ATP-cones are at the bottom of the figure. (**b**) View rotated by 90° around a horizontal axis relative to (**a**) and thus viewed from the bottom, showing the ATP-cones. (**c, d**) Cartoon representation of the tetramer from the same angles as panels (**a**) and (**b**), respectively, and with the same chain colouring. The twofold symmetry axis relating the two dimers of the tetramer is shown as a yellow line. The 12 dATP molecules in the tetramer are shown as sticks. The dimer axis of the left-hand dimer's core domains is shown as a dark blue line and the dimer axis of the ATP-cone pair as a black line. (**e**) Details of the interaction of dATP molecules at the allosteric activity site in the ATP-cone. (**f**) Closeup of the interaction area between the roof of the ATP-cone of one dimer and the core domain of the other dimer (marked with a black box in panel d). The most significant residues are labelled.

The online version of this article includes the following figure supplement(s) for figure 9:

**Figure supplement 1.** Cryo-EM data processing workflow for PcNrdD in the presence of dATP.

**Figure supplement 2.** The ATP-cones, linker, entire glycyl radical domain (GRD), and NxN flap are all disordered in both monomers of the dATP-bound PcNrdD dimer.

**Figure supplement 3.** ATP and dATP bind differently to the PcNrdD ATP-cone.

**Figure supplement 4.** The linker region between the ATP-cone and the first helix of the core domain interacts very differently in the two monomers of each dimer.

**Figure supplement 5.** Cryo-EM data processing workflow for PcNrdD in the presence of dATP/CTP.

**Table 4.** Structural similarity comparisons to PcNrdD ATP-cone alone.
The PcNrdD ATP-cone in the comparison is the one from the dATP tetramer structure.

| PDB ID | Protein | No. of bound nucleotides | Z-score | Residues aligned | RMSD (Å) | Sequence identity (%) |
|--------|---------|--------------------------|---------|------------------|----------|------------------------|
| 5IM3 | *P. aeruginosa* NrdA | 2 dATP | 12.2 | 92 | 1.9 | 29 |
| 5OLK | *L. blandensis* NrdB | 2 dATP | 12.1 | 93 | 1.7 | 33 |
| 7P37 | *S. coelicolor* NrdR | 2 ATP | 10.0 | 82 | 2.4 | 17 |
| 6AUI | Human NrdA/dATP | 1 dATP | 9.5 | 88 | 2.8 | 19 |
| 7AGJ | *A. aeolicus* NrdA/ATP | 2 ATP | 8.7 | 88 | 2.2 | 17 |
| 5R1R | *E. coli* NrdA E441A | – | 8.2 | 88 | 3.3 | 16 |
| 7MDI | *N. gonorrhoeae* NrdA | 1 dATP | 7.4 | 87 | 3.4 | 16 |

dATP-loaded ATP-cones of the RNR class I active site subunit from PaNrdA (PDB ID 5IM3) and the class I radical generating subunit from LbNrdB (5OLK) to which the PcNrdD ATP-cones have 29% and 33% sequence identity, respectively (*Table 4*). The DALI server reveals an rmsd in Cα positions of 1.9 and 1.7 Å, respectively. The next most similar ATP-cone is that of the RNR transcriptional regulator NrdR from *Streptomyces coelicolor* in its dodecameric form (*Rozman Grinberg et al., 2022*) binding two ATP molecules, with 82 aligned residues, rmsd 2.4 Å and sequence identity 17%.

The two dATP molecules bind with their negatively charged triphosphate moieties oriented towards each other (*Figure 9c*). No Mg$^{2+}$ ion is observed between the triphosphate tails, but this may be an artefact of the cryo-EM method, as the PaNrdA and LbNrdB crystal structures suggest that this ion is necessary for charge neutralisation. As in the ATP-bound form, further charge compensation is achieved by extensive coordination by, or proximity of, basic side chains, for example Lys7, Arg8, Arg87, and Arg91. As previously observed in PaNrdA and LbNrdB, the adenosine base of the 'inner' dATP (dATP1) fits into a pocket under the β-hairpin roof of the ATP-cone and the base of the 'outer' nucleotide (dATP2) is sandwiched between an aromatic residue (Tyr83) and the side of the first α-helix in the domain. No direct contacts are seen between the dATP molecules bound to the ATP-cone and residues in the core domain.

*Figure 9—figure supplement 3* shows the differences in binding modes of dATP in the dATP-bound tetramer and ATP in the ATP–CTP dimer. The high mobility of the ATP-cone in the latter precludes a detailed analysis. However, it is clear that the triphosphate tails have very different orientations. In the ATP complex, the tail of ATP2 is oriented away from the C-terminal helix towards the base of ATP1, such that the β-phosphate moiety may even H-bond to the base of ATP1. The side chain of Arg87 fills the space vacated by the γ-phosphate of ATP2. Instead of H-bonding to the γ-phosphate of ATP2, Arg91 swings away and interacts with the equivalent moiety of ATP2 in the ATP-cone of the other chain. The structures suggest that the additional 2'-OH group in ATP may cause conformational changes in the ribose ring that propagate to the phosphate tail, but further analysis requires improved resolution.

As in PaNrdA and LbNrdB, the ATP-cones in the dATP-bound tetramer self-interact through a pair of α-helices at the C-terminal end of the ATP-cone. Remarkably, the pair of ATP-cones is highly asymmetrically positioned with respect to the twofold axis of its core dimer, being displaced as a rigid body towards the other dimer (*Figure 9b*). Both pairs of ATP-cones are displaced in the same way, giving the complex overall C2 symmetry around an axis bisecting the pair of dimers. Consequently, the twofold axes of the ATP-cones and the core dimer are not aligned (*Figure 9b*). The asymmetric arrangement is associated with two distinct conformations of the linker 93–103 between the fourth helix of the ATP-cone (*Figure 9—figure supplement 4*) and the core domain. In chains A and C, a small kink leads from helix 4 into a short helical segment that packs antiparallel to helix 122–142 in the core domain. In contrast, in chains B and D, the short helix is parallel to helix 122–142 and is joined to helix 4 at a 45° angle. The rest of the polypeptide between residues 103 and 121, including the NxN flap, is disordered in both linkers.

Interestingly, all contacts between the two dimers of the tetramer are mediated by interactions between one of the ATP-cones in one dimer (B or D) and one of the core domains of the other dimer (C or A, respectively). *Figure 9d* shows the interactions of the roof of the ATP-cone with two outer helices from the C-terminal half of the barrel domain (residues 458–485 and 550–569, preceding the sixth and seventh strands). The interactions are mostly hydrophobic but are reinforced by several H-bonds, for example between Gln3D–Gln458A, Ser53D–Gln458A, Arg11D–Asp468A, the main chain amide of Ile12D and Tyr557A. The amount of buried surface area is small: 717 Å$^2$, or around 1% of the total surface area of each dimer. For comparison, the monomer–monomer interactions within each dimer bury 11.7% of the total area. The residues on the core domain of PcNrdD involved in interactions with the ATP-cone of the other dimer do not show high sequence conservation, even when the alignments are restricted to the most similar sequences.

## Cryo-EM structures of PcNrdD–dATP complexes produced in the presence of CTP

To investigate whether substrate had any allosteric effect on the conformations of the ATP-cone, flap, and GRD, we determined separate cryo-EM structures of tetrameric and dimeric PcNrdD, from the same grid, in the presence of 0.5 mM dATP and 0.5 mM CTP, achieving a resolution of 2.6 Å from 1,105,348 particles before post-processing for the tetramer and 2.6 Å from 1,132,695 particles for the dimer (*Figure 9—figure supplement 5*). Both forms were almost identical to those seen in the presence of dATP alone, and no CTP could be seen bound to the active site (*Figure 6—figure supplement 3*). This strongly suggests that substrate binding is dependent on ordering of the flap region by binding of ATP to the ATP-cone.

## Hydrogen–deuterium exchange mass spectrometry

To validate our structural and biochemical hypotheses, we further examined the degree of order of the different structural elements of PcNrdD in solution using hydrogen–deuterium exchange mass spectrometry (HDX-MS). This method measures the degree of protection of peptides from H/D exchange in a deuterated solution, which reports on how exposed they are. Three different samples were analysed: apo-PcNrdD and the complexes with dATP–CTP and ATP–CTP.

The apo form shows high deuterium uptake over the entire ATP-cone, linker, and NxN flap, from residues 1 to 117 (*Figure 10a*). Analysis of differences in uptake in the N-terminal region between the apo protein and the ATP or dATP complexes clearly validates binding of both nucleotides to the ATP-cone (*Figure 10b*). Peptides spanning residues 2–84 exhibit clear protection from HDX in both nucleotide-bound states. Furthermore, bimodal deuterium uptake (*James et al., 2022*) was observed for N-terminal peptides spanning residues 2–23 and 83–99. For peptides very close to the N-terminus (2–23), the apo form only exhibits one high uptake (dynamic) conformational state while for the two ligand bound states there was a clear bimodal behaviour with one high uptake and one protected state of an approximately equal ratio (*Figure 10c* and *Figure 10—source data 1*). In-depth analysis of the causes is difficult, but possible interpretations are that the bimodality is due to a mixture of occupied and empty ATP-cones, or more likely to the mixture of dimers and tetramers.

Interestingly, peptides spanning amino acids 81–99 in the linker region have a bimodal uptake that is most pronounced in the apo state. For the nucleotide-bound forms there are also two populations that converge into a single state at 3000 s (*Figure 10c*). This suggests a mixture of conformational states with flexible and locked linkers, consistent with the highly dynamic nature of residues 85–100 observed in the cryo-EM structures, where in the dATP-bound tetramer, the linker is ordered to residue ~100, while in the dimer the ATP-cone and linker are highly dynamic and the first ordered structure is observed from residue 120.

The NxN flap at residues 101–117 has a high H/D exchange rate already at the shortest labelling time, with no change at longer times for any state. While surprising in the context of the cryo-EM observations, this might be the result of very fast dynamics in the NxN flap. For the regions 118–126, a higher exchange rate in the dATP form than in the ATP form at the longest time point, most likely because substrate is not bound to the active site.

Clear differences between the ATP and dATP complexes are seen in the regions 178–204, which contains the β-hairpin of the s-site (*Figure 10c*, panels 178–204) that is highly protected with dATP. This is consistent with the structures that show no ATP bound at the s-site but full occupancy of dATP.

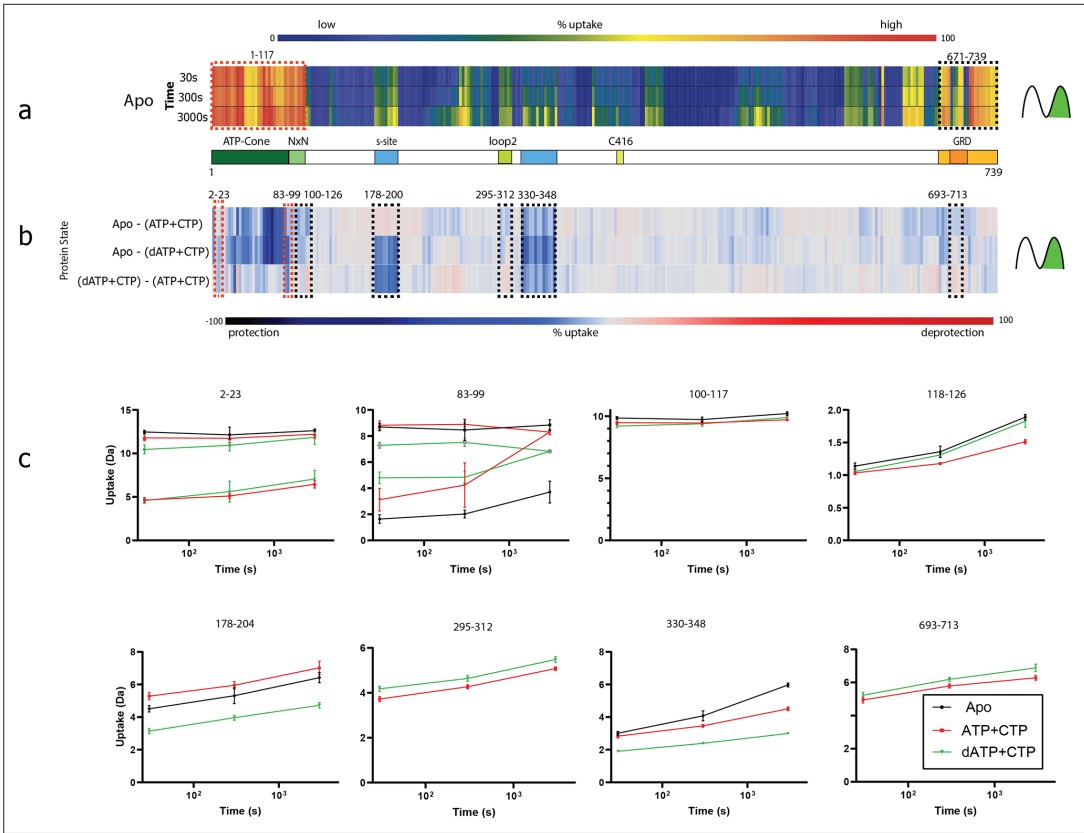

**Figure 10.** Summary of the hydrogen–deuterium exchange mass spectrometry (HDX-MS) experiments. (**a**) Heat map showing the deuterium uptake of the apo state for time points 30, 300, and 3000 s. Warm colours represent fast uptake and cold colours a low uptake. Both the N- and the C-terminal regions have a high uptake already at 30 s. For the N-terminal bimodal uptake, only the fast-exchanging population (rightmost) is depicted. The influence of bimodal uptake and more detail can be seen in *Figure 10—source data 1*. (**b**) Differential chiclet plot (*Zhang et al., 2021*) collapses the timepoints into a single maximum deuterium uptake difference for states apo-(ATP + CTP), apo-(dATP + CTP), and (dATP + CTP)–(ATP + CTP). Colours range from deep blue for a high degree of relative protection to deep red for high deprotection. Areas marked with dashed red boxes are peptides showing bimodal uptake, that is two conformational populations present simultaneously, and black dashed boxes denote areas with peptides having EX2 exchange (only one conformational population in the observed timespan). (**c**) Deuterium uptake plots from dashed areas in (**b**). For peptides 2–23 and 83–99, when there are two lines with the same colour, the top traces are for the fast-exchanging population and the bottom for a less dynamic/more protected population. Note that for peptides 83–99, the apo state has two populations at all timepoints, while the ATP and dATP converge at 3000 s to one (dynamic) population. For peptides 295–312 and 693–713, the apo curve has been removed for visual clarity.

The online version of this article includes the following source data for figure 10:

**Source data 1.** Detailed analyses of the hydrogen–deuterium exchange mass spectrometry (HDX-MS) data, including the bimodal uptake distribution in the N-terminal region.

Peptides between 330 and 348, which are in contact with the β-hairpin, are also protected in the dATP complex. The regions 295–312, that is loop 2, are deprotected in the dATP form relative to ATP. This is also in agreement with the cryo-EM structures, in which loop 2 is partially disordered in the dATP complex but ordered in the ATP complex in the monomer with bound CTP.

The HDX-MS results show a high degree of deuterium uptake in the GRD in all forms, indicating a high degree of mobility. This is consistent with the cryo-EM structures, in which the GRD overall has *B*-factors higher than the core domain (though the glycyl-radical containing loop has similar *B*-factors to the core, being buried). From the HDX-MS data, it is clear that the GRD is highly dynamic, and while a trend of increased dynamics for some C-terminal peptides in the GRD can be seen for the

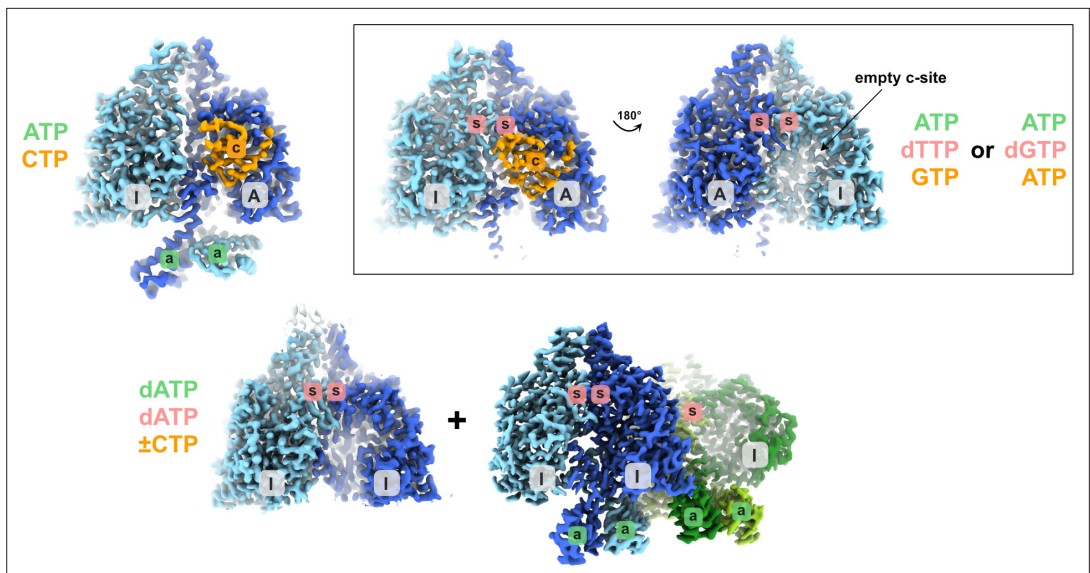

**Figure 11.** Schematic showing the different structural states of PcNrdD described in this work. The glycyl radical domain (GRD) is coloured orange where present. Inactive and active monomers are labelled I and A, respectively. The a-site in the ATP-cone is labelled with 'a' on a green background, the s-site at the dimer interface with 's' on a red background and the active site with 'c' on an orange background. Labels are only present when a nucleotide is observed in the structure. The ligands bound are listed, coloured according to the sites they bind to. In the dATP/dATP–CTP tetramer, one of the inactive monomers and its s-site are hidden behind the dimer in the foreground.

dATP–CTP state, a detailed analysis is confounded by the limited time resolution and the complexity of the sample, which consists of a mixture of conformations.

## Discussion

RNR is an essential enzyme for all free-living organisms. Its complex regulation at the levels of transcription, overall activity and substrate specificity makes it quite unique in Nature (*Mathews, 2016*; *Mathews, 2018*). A fundamental understanding of this regulation could for example pave the way for better antibiotics. The ATP-cone is a small, genetically mobile domain frequently found at the N-terminus of RNR catalytic subunits of all major classes, as well as in the bacterial RNR-specific transcription factor NrdR (*Rozman Grinberg et al., 2022*). Its role in allosteric regulation of class I aerobic RNRs has been extensively studied. In the presence of dATP, the ATP-cone mediates the formation of a surprising variety of different oligomeric complexes that nevertheless have in common that they block the formation of an (observed or inferred) active complex between the catalytic and radical generating subunits that permits proton-coupled electron transfer on each catalytic cycle (reviewed in *Martínez-Carranza et al., 2020*).

Despite their catalytic subunits being evolutionarily related to those of class I, anaerobic class III RNRs have a fundamentally different activation mechanism that is likely common to all GREs (*Backman et al., 2017*; *Lundahl et al., 2022*). Once generated on the catalytic subunit by encounter of the reductase with its radical-SAM family activase, the glycyl radical can catalyse hundreds of cycles of turnover before being exhausted (*Torrents et al., 2001*). Allosteric inhibition could thus in principle proceed by two mechanisms: by blocking the initial encounter of the reductase and activase, or by preventing subsequent transfer of the radical to the substrate. The first of these hypotheses is intuitively more similar to the steric blocking mechanisms encountered in class I RNRs; the latter is more challenging to understand, as the glycine and cysteine are buried in close proximity in the active site.

In this work, we present the first biochemical, biophysical, and structural characterisation of an anaerobic, class III RNR containing an ATP-cone, providing the first picture of the structural basis for allosteric activity regulation in this large family. We show that the ATP-cone can bind two molecules of ATP or dATP, but not both simultaneously. HDX-MS confirms that binding of either nucleotide reduces the exposure of the ATP-cone to deuterium exchange and supports the biophysical findings of dATP

binding to the s-site. As expected, the presence of an ATP-cone confers inhibition of enzyme activity at high dATP concentrations. GEMMA experiments show that PcNrdD exists in a mixture of dimeric and tetrameric states under most experimental conditions studied, with the equilibrium shifted towards dimers in the presence of ATP. Addition of inhibiting concentrations of dATP shifts the equilibrium towards tetramers. The equilibria are not affected by substrate or specificity effector. EPR spectroscopy clearly shows that the glycyl radical is formed even in the presence of dATP. Together, these results show that dATP inhibition in class III RNRs does not occur by blocking the initial encounter of reductase and activase. Instead, our nucleotide-binding results and structures show that dATP inhibition prevents binding of substrates.

The seven cryo-EM structures of PcNrdD in a variety of nucleotide complexes presented here suggest a mechanism in which binding of ATP or dATP modulates activity by affecting the dynamics of a tightly knit network of structural elements including the critical GRD in the C-terminal region of the enzyme. A summary of the conformational states is shown in *Figure 11*. In the presence of ATP and substrate CTP, the GRD is fully ordered in one monomer and Gly711 is found at its expected position proximal to Cys416, to which it will deliver its radical to initiate each catalytic cycle and accept it back at the end of the cycle. The CTP substrate is bound in this monomer and a conserved but previously unobserved flap containing two Asn residues (NxN flap) forms over the top of the substrate, making H-bonding interactions with it. Loop 2, which mediates between the substrate specificity site and the active site, is fully ordered. In contrast, in the other monomer, no substrate is bound, loop 2 is disordered and the flap and C-terminus are more dynamic. At the same time, the ATP-cones themselves are relatively flexible. This is not unexpected for this type of ATP-cone, which binds two nucleotides and was first observed in the class I RNR PaNrdA (*Johansson et al., 2016*). SAXS studies showed that the ATP-cone in PaNrdA is flexible in the presence of ATP, while in its dATP-bound form it forms a symmetric interaction with an ATP-cone on another dimer, causing the enzyme to tetramerise. A similar behaviour was seen even when this type of ATP-cone was evolutionarily 'transplanted' to the N-terminus of the radical generating subunit, as seen in LbNrdB (*Rozman Grinberg et al., 2018a*).

By careful isolation of a subset of particles, we were able to characterise an unusual dimer of ATP-cones in the ATP–CTP complex, where four ATP molecules bind with all their triphosphate tails pointing towards each other. Such an orientation is unprecedented for RNRs, but such tail-to-tail orientations, compensated by extensive positive charges, are found in for example adenylate kinases (*Berry et al., 1994*). The relevance of this arrangement remains to be elucidated. Interestingly, when dTTP was added to the s-site in the ATP–dTTP–CTP complex, the ATP-cones became even more dynamic and could only be observed in one monomer of the dimer as diffuse density. This behaviour was reproduced with dGTP in the s-site. Both complexes with s-site effector are functionally more relevant forms than the ATP–CTP complex, as in the cell, a form with empty s-site is unlikely to exist. The partitioning of monomers into active and inactive is amplified by the presence of s-site effector (dTTP or dGTP) in both complexes with ATP. Only one of the active sites is occupied by substrate, with ordered NxN flap and C-terminal GRD. This is correlated with two distinct conformations of loop 2 in the two monomers. In the inactive monomer, the NxN flap and GRD are disordered, as loop 2 blocks their active conformation and concomitantly substrate binding. These results point to only one of the PcNrdD monomers being active at a given time and also to a significant cross-talk between the a-site, s-site, and active site. This is corroborated by the finding that an s-site effector is necessary for retention of ATP by the ATP-cone when PcNrdD is first incubated with these nucleotides then purified without them.

In the presence of inhibitory concentrations of dATP we were able to isolate structures of both dimers and tetramers from the same cryo-EM sample, in agreement with the GEMMA results that show a shift towards the tetrameric form. In both forms, the C-terminal GRD is highly mobile in all monomers such that it cannot be seen in the cryo-EM reconstructions. An exposed glycyl radical is incompatible with its stability, so it is possible that the dynamics of the GRD are exaggerated somewhat by the vitrification process. Consistent with this, the HDX-MS results show a small but noticeable deprotection of peptides from the GRD in the presence of dATP–CTP compared to ATP–CTP.

In the tetramers, pairs of ATP-cones asymmetrically disposed relative to the twofold axes of their respective PcNrdD dimers associate through the 'roof' of the 'inner' dATP-binding site to the core domain of the other dimer. While this allowed us to build complete models for the ATP-cones and analyse dATP binding, the role of tetramerisation in the inhibition mechanism is unclear. The tetrameric

structures appear to be facilitated by a dATP-induced 'stiffening' of the pair of ATP-cones, similar to what was seen in PaNrdA and LbNrdB, and their interaction with a second dimer of PcNrdD, but the interaction area is small (1% of the total surface area) and the residues involved are not highly conserved in NrdDs, even from the most similar sequences. Therefore, further studies of NrdDs containing ATP-cones are required in order to determine whether oligomerisation is a general phenomenon in the family. Furthermore, the GRD is mobile in all complexes with dATP in the a-site, whether dimeric or tetrameric. Since both the ATP-cone and the GRD are disordered in the dATP-bound dimers, the exact molecular mechanism by which dATP induces disorder of the GRD remains somewhat elusive.

Taken together, our results show a clear correlation between ATP or dATP binding to the ATP-cone and activity or inhibition, respectively. The dynamical transitions involve a previously uncharacterised but highly conserved structural element, the NxN flap, that folds over the substrate and furthermore acts as a structural bridge between the ATP-cone and the C-terminal GRD. An increase in dynamics of these elements prevents substrate binding and thus transfer of the radical from the GRD to the substrate. Interestingly, a similar flap is formed over the top of the active site in the *E. coli* active complex, but it is formed by part of the C-terminal region of the separate radical generating subunit (*Kang et al., 2020*). Thus, a locking of the substrate in the active site appears to be a conserved feature of active class I and III RNRs. However, in class I RNR there is no tunnel allowing substrate access to the active site in the locked state. This is consistent with the fact that the class I subunits dissociate on each catalytic cycle, while in class III the glycyl radical can catalyse many substrate reductions before having to be regenerated. It remains enigmatic exactly how the tiny chemical difference between dATP and ATP (the 2′-OH group) results in such major conformational changes, as in the ATP–CTP complex, where the ATP-cones are visible, the a-site in the ATP-cone is separated from the NxN flap by at least 30 Å and in the most biologically relevant complexes with additional s-site effector, the ATP-cone is very disordered. Nevertheless, the present results give first insights into allosteric activity regulation in anaerobic RNRs and add yet another aspect to the surprisingly wide range of allosteric conformational changes that can be mediated by the binding of the two highly similar nucleotides ATP and dATP to a small, evolutionarily mobile protein domain.

## Materials and methods
### Cloning of *nrdD* and *nrdG* from *P. copri*

The genes encoding the reductase (NrdD) and activase (NrdG) proteins from *P. copri* were synthesised by GenScript with codon optimisation for *E. coli* and subcloned into the pBG102 plasmid (pET27 derivative) (Center for Structural Biology, Vanderbilt University) between the BamHI and EcoRI restriction sites to produce His6–SUMO–NrdD and His6–SUMO–NrdG protein constructs.

### Overexpression and purification of PcNrdD and PcNrdG

Plasmids containing the *nrdD* gene or the *nrdG* gene were transformed into *E. coli* BL21 (DE3) star and *E. coli* BL21 (DE3) competent cells, respectively. Cells (6 l) were grown at 37°C in Luria Broth supplemented with kanamycin (50 µg/ml) to an $A_{600}$ of 1.2. Protein expression was then induced with 0.5 mM isopropyl ß-D-1-thiogalactopyranoside (IPTG) and incubation was extended overnight at 20°C. After centrifugation, pellets were suspended in 60 ml of lysis buffer 1 a (50 mM Tris–HCl pH 8, 500 mM KCl, 0.5 mM tris(2-carboxyethyl)phosphine [TCEP]) for PcNrdD and lysis buffer 2a (50 mM Tris–HCl pH 8, 500 mM NaCl, 0.5 mM TCEP) for PcNrdG, supplemented with lysozyme (0.5 mg/ml), and disrupted by sonication. Cell debris was removed by ultracentrifugation at 210,000 × *g* for 1 hr at 4°C. The supernatant was then loaded on an immobilised metal affinity Ni-NTA column (HisTrap 5 ml; Cytiva) equilibrated with buffer 1b (50 mM Tris–HCl, pH 8, 300 mM KCl) or buffer 2b (50 mM Tris–HCl, pH 8, 300 mM NaCl), supplemented with 50 mM imidazole as appropriate. The column was washed extensively with the corresponding buffer containing 50 mM imidazole and the His6-SUMO tagged proteins were eluted using 300 mM imidazole. The proteins were collected and dialysed overnight at 4°C against buffer 1b or 2b supplemented with 1 mM 1,4-dithiothreitol (DTT) in the presence of the PreScission Protease (150 µM) to cleave the affinity-solubility tag. The GST-3C-protease (PreScission) was expressed using pGEX-2T recombinant plasmids. After induction at 25°C with 0.1 mM IPTG for 20 hr, the PreScission Protease was purified using glutathione–Sepharose chromatography. After the dialysis, the cleaved PcNrdD and PcNrdG proteins were centrifuged at 4°C for 10 min and loaded

onto a HisTrap column equilibrated with buffer 1b or 2b, supplemented with 50 mM imidazole and the flow through containing the cleaved protein was collected. Following cleavage of the His$_6$-SUMO tag, the proteins incorporated a non-native N-terminal Gly-Pro-Gly-Ser sequence. The purified preparations are called as-purified PcNrdD and PcNrdG, respectively.

A fraction of as-purified PcNrdD was precipitated by addition of HCl, then centrifuged at 15,000 × $g$ for 5 min. UV–visible spectra of the supernatant were recorded before and after protein precipitation to estimate the amount of nucleotide contamination. PcNrdD preparations with ≤5% nucleotide contamination are referred to as apo-PcNrdD.

Fractions containing apo-PcNrdD and as-purified PcNrdG were concentrated and loaded on a gel filtration column (Hiload 16/60 Superdex S200; Cytiva) in 25 mM Tris–HCl, pH 8, 250 mM NaCl, and 5 mM DTT. The purified protein was concentrated to 20 mg/ml with an Amicon ultrafiltration device (100 kDa cutoff for NrdD and 10 kDa cutoff for NrdG; Millipore), frozen in liquid nitrogen, and stored at −80°C.

## [Fe–S] cluster reconstitution and purification of holo-PcNrdG

The reconstitution of the [4Fe–4S] cluster and purification of PcNrdG containing an iron–sulfur cluster (holo-PcNrdG) were performed under strict anaerobic conditions in an Mbraun glove box kept at 18°C and containing less than 0.5 ppm O$_2$. PcNrdG was treated with 5 mM DTT for 10 min then incubated for 3 hr with a fivefold molar excess of ferrous ammonium sulphate and L-cysteine in the presence of 2 µM $E.\ coli$ cysteine desulfurase CsdA. The holo-PcNrdG was then loaded onto a Superdex 200 10/300 gel filtration column (Cytiva) equilibrated in 25 mM Tris–HCl, 250 mM NaCl, and 5 mM DTT (**Figure 1—figure supplement 2**). The peak containing the soluble holo-PcNrdG was then concentrated to 10 mg/ml on a Vivaspin concentrator (10 kDa cutoff).

## In vitro enzymatic assay and HPLC analysis

Activity assays were performed under strict anaerobic conditions inside a glovebox in 25 mM Tris pH 8, 50 mM KCl, 10 mM DTT in a volume of 100 µl. A standard pre-reaction mixture contained 1 µM apo-PcNrdD, 2.5 µM holo-PcNrdG, 500 µM S-adenosyl-methionine (SAM), 5 mM MgCl$_2$, 1 mM GTP (or 1 mM CTP), and different allosteric effectors dTTP (1 mM), ATP (0–5 mM), or dATP (0–1 mM). The pre-reaction mixture was incubated for 5 min at 37°C before adding simultaneously 10 mM sodium formate and sodium dithionite (12.5 molar excess). When GTP was used as substrate, dTTP was used as s-site effector, and ATP or dATP were titrated to the ATP-cone (a-site). When CTP was used as substrate the s-site was empty at the beginning of the titration and ATP or dATP can plausibly bind both to the s-site and to the a-site in these experiments.

The reaction was incubated for 10 min at 37°C and stopped by the addition of 2.5 µl of 3 M formic acid (FA). Product formation was analysed by HPLC using an Agilent ZORBAX RR StableBond (C18, 4.6 × 150 mm, 3.5 µm pore size) equilibrated with buffer A (10% methanol, 50 mM potassium phosphate buffer, pH 7, 10 mM tetrabutylammonium hydroxide). Samples of 10 µl were injected and eluted at 0.5 ml/min with a step isocratic flow of 40–100% buffer B (30% methanol, 50 mM potassium phosphate buffer, pH 7, 10 mM tetrabutylammonium hydroxide). Compound identification and product quantification were performed by external calibration and NrdD-specific activity was determined.

From a direct plot of activity versus concentration of effector, the $K_L$ values (the concentration of an allosteric effector that gives half maximal enzyme activity) for binding of effectors to the a-site, were calculated in KaleidaGraph using the equation:

$$V = V_{max} \times [NTP/dNTP]/(K_L + [NTP/dNTP]) \tag{1}$$

$K_i$ for non-competitive dATP inhibition of GTP reduction was calculated in KaleidaGraph using the equation:

$$V = V_{max}/(1 + [dNTP]/K_i) \tag{2}$$

and $K_L$ and $K_i$ for dATP-dependent activation and non-competitive dATP inhibition of CTP reduction were calculated in KaleidaGraph using the equation:

$$v = (V_{max} \times [dNTP]/(K_L + [dNTP])) + (V_{inh}/(1 + [dNTP]/K_i)) \tag{3}$$

## ITC experiments

ITC experiments were carried out at 20°C on a MicroCal ITC-200 system (Malvern Instruments Ltd) in ITC buffer containing 25 mM 2-[4-(2-hydroxyethyl)piperazin-1-yl]ethanesulfonic acid (HEPES), pH 7.5, 100 mM KCl, 10 mM $MgCl_2$, and 0.5 mM TCEP with a stirring speed of 1000 rpm. An initial injection volume was 0.6 µl over a duration of 1.2 s. ApoPcNrdD was prepared in the ITC buffer using SEC, followed by the addition of 1 mM dTTP and 1 mM GTP and incubation at room temperature for 5 min prior to loading to the cell. ATP and dATP ligands were directly diluted in the ITC buffer. Titration of ligand into buffer was also performed to control for heats of dilution and/or buffer mismatch. For dATP-binding analysis, the concentration of PcNrdD in the cell was 50–100 µM and dATP concentration in the syringe 0.6–1 mM. For ATP titration into PcNrdD, protein concentration in the cell was 200 µM and ATP concentration in the syringe was 0.6–1.5 mM. All subsequent injection volumes were 1.7–2.5 µl over 3.4–5 s with a spacing of 160 s between the injections. Data for the initial injection were not considered. The data were analysed using the one set of sites model of the MicroCal ITC-200 analysis software (Malvern Panalytical). Standard deviations in thermodynamic parameters, $N$ and $K_D$ were estimated from the fits of three different titrations.

## Microscale thermophoresis

Both binding of nucleotides to the ATP-cone in PcNrdD and binding of substrates CTP and GTP to PcNrdD were assessed using MST. PcNrdD was labelled using Monolith Protein Labeling Kit RED-NHS 2nd generation according to the manufacturer's protocol. MST was performed using the Monolith NT.115 instrument (NanoTemper Technologies GmbH) at room temperature. Binding of GTP and CTP was assayed in MST buffer containing 40 mM HEPES pH 7, 50 mM KCl, 50 and 100 mM $MgCl_2$, 5 mM DTT, 0.1% Tween-20, 1 mM dTTP (only for GTP binding), and either 5 mM ATP or 1 mM dATP. The 16 reaction tubes were prepared by the addition of 2 µl MST buffer (concentrated five times), 5 µl GTP or CTP of the desired concentration and 1 µl RED dye-labelled NrdD in a total volume of 10 µl. Final NrdD concentration in the binding reaction was 11 nM while the binding partner concentrations were between 70 mM and 2 µM for CTP and 50 mM and 1.5 µM for GTP. Binding of ATP and dATP to the ATP-cone was assayed in MST buffer containing 25 mM HEPES pH 7.5, 100 mM KCl, 10 mM $MgCl_2$, 0.5 mM TCEP, 0.1% Tween-20, 1 mM dTTP, and 5 mM GTP. A 16-step dilution series of the binding partners was prepared by adding 5 µl buffer to 15 tubes. ATP or dATP was added to the first tube to the final concentration of 4 mM and 5 µl was transferred to the second tube and mixed by pipetting (1:1 dilution series). To each tube of the dilution series 5 µl of PcNrdD was added to reach a final concentration of 11 nM. The samples were transferred to Monolith NT.115 Series Premium Capillaries (NanoTemper Technologies GmbH), which were scanned using the MST instrument (100% excitation power, medium MST power). Obtained MST data were analysed and fitted using the MO. Affinity Analysis v2.3 software (NanoTemper Technologies) with default parameters. $K_D$ and standard deviation for GTP and CTP binding to PcNrdD in the presence of ATP and for ATP and dATP binding to the ATP-cone were calculated using fits from at least three individual titrations. $K_D$s for CTP or GTP binding to PcNrdD in the presence of dATP were extremely high and could not be reliably determined since the titration curves did not reach a plateau and therefore are only estimates.

## Nucleotide binding to the PcNrdD ATP-cone

PcNrdD was incubated with nucleotides at room temperature for 30 min and then desalted on a NAP-5 column. The desalted protein was boiled for 10 min, centrifuged, and the supernatant was loaded on a Agilent ZORBAX RR StableBond (C18, 4.6 × 150 mm, 3.5 µm pore size) HPLC column to evaluate the amount of nucleotides retained by the protein. Samples of 100 µl were injected on the column equilibrated with buffer A (10% methanol, 50 mM potassium phosphate buffer, pH 7, 10 mM tetrabutylammonium hydroxide) and eluted at 1 ml/min with a gradient of 40–100% buffer B (30% methanol, 50 mM potassium phosphate buffer, pH 7, 10 mM tetrabutylammonium hydroxide). Compound identification and product quantification were performed by comparison with injected standards. Three different sets of experiments were performed, each set consisting of incubation with only ATP, only dATP, and a combination of ATP and dATP. In the first set of experiments 75 µM apo-NrdD was incubated with 1 mM ATP or 1 mM dATP, and when 1 mM each of ATP and dATP was used, dATP was added 15 min after addition of ATP. After 30-min incubation the mixtures were desalted in a buffer without nucleotides. In the second set of experiments 75 µM apo-NrdD in presence of

1 mM dTTP and 5 mM GTP was incubated with 3 mM ATP or 1 mM dATP, and when both nucleotides were combined dATP was added 15 min after addition of ATP. After 30-min incubation the mixtures were desalted in a buffer without nucleotides. In the third set of experiments 75 µM apo-NrdD in presence of 2 mM dTTP and 5 mM GTP was incubated with 3 mM ATP or 1 mM dATP, and when both nucleotides were combined dATP was added 15 min after addition of ATP. After 30-min incubation the mixtures were desalted in a buffer containing 0.1 mM dTTP and 1 mM GTP. It was possible to separate all phosphorylation levels of adenosine and deoxyadenosine nucleotides in the first set of experiments, but not in the second and third sets due to overlapping peaks of the s-site and substrate nucleotides (*Table 1*).

## Gas-phase electrophoretic mobility molecular analysis

The GEMMA instrumental setup and general procedures were as described previously (*Kaufman et al., 1996*; *Rofougaran et al., 2008*). Apo-PcNrdD was equilibrated into a buffer containing 100 mM ammonium acetate pH 7.8, then applied onto a Sephadex G-25 chromatography column. In addition, 5 mM DTT was added to the PcNrdD protein solution to increase the protein stability. Prior to analysis, the protein sample was diluted to 2 µM in a buffer containing 100 mM ammonium acetate, pH 7.8, 0.005% (vol/vol) Tween 20, magnesium acetate (equimolar to the total nucleotide concentration) and the corresponding nucleotides (where indicated). The protein samples were then incubated for 2 min at room temperature, centrifuged and applied to the GEMMA instrument for data collection. Each sample was scanned five times and the signal was added together to obtain the traces presented. The experiments were performed with a flow rate driven by 2 psi to minimise noise signals that may appear with elevated nucleotide or protein concentrations at the manufacturer's recommended flow rate driven by 3.7 psi.

## Glycyl radical characterisation by EPR

EPR samples were handled under strictly anaerobic conditions inside a glovebox in 25 mM Tris–HCl pH 8, 50 mM KCl, 5 mM DTT in a final volume of 100 µl. A standard pre-reaction mixture containing 75 µM holo-PcNrdG, 1 mM *S*-adenosyl-methionine, was treated with a 12.5 excess of dithionite and was incubated for 20 min at 37°C. The pre-mixture was added to a mixture of 50 µM apo-PcNrdD, 10 mM sodium formate, supplemented or not with 1.5 mM ATP or 1 mM dATP and in the presence of 1 mM CTP. The final reaction was incubated for 20 min at 37°C and 100 µl sample was transferred into a standard EPR tube (Wilmad LabGlass 707-SQ-250M) and stored in liquid nitrogen before recording the EPR spectrum.

X-band CW-EPR spectra of PcNrdD were recorded on a Bruker ELEXYS-E500 spectrometer operating at microwave frequency of 9.38 GHz, equipped with a SuperX EPR049 microwave bridge and a cylindrical $TE_{011}$ ER 4122SHQE cavity in connection with an Oxford Instruments continuous flow cryostat. Measuring temperature at 30 K was achieved using liquid helium flow through an ITC 503 temperature controller (Oxford Instruments). For achieving non-saturating conditions, a microwave power of 10 µW was applied for all measurements. Other EPR settings were: modulation frequency of 100 kHz; modulation amplitude of 8 G; and accumulation of 16 scans for an optimal S/N ratio. Double integrated spectra were used for intensity analysis.

## Cryo-EM sample preparation and data acquisition

The proteins for cryo-EM analysis were all prepared in the same manner. Apo-PcNrdD was incubated with 1 mM NTP or dNTP at room temperature for 5 min, then centrifuged before injection onto a Superdex S200 10/300 column pre-equilibrated with a buffer containing 25 mM HEPES–NaOH pH 7.5, 100 mM KCl, 0.5 mM TCEP, 10 mM $MgCl_2$, and supplemented with varying combinations of 0.5 mM NTP or 0.2 mM dNTP. The relevant peak was collected, then diluted at 0.5 mg/ml and the corresponding nucleotides were added to different final concentrations (0.5–5 mM NTP or 0.5–1 mM dNTP) (*Table 5*).

The grids were glow-discharged for 60 s at 20 mA using a GloQube (Quorum) instrument, then prepared in the following manner: 3 µl of sample at 0.5 mg/ml were applied on Quantifoil 1.2, 1.3 300 mesh Cu grids or Quantifoil 2.1, 300 mesh Au grids and plunge-frozen in liquid nitrogen-cooled liquid ethane using a Vitrobot Mark IV (Thermo Fisher Scientific) with a blot force of 1 and followed by 5 s blot time, at 4°C and 100% humidity. The grids used for each sample are specified in *Table 5*.

**Table 5.** Data collection, processing, and refinement statistics for the cryo-EM structures of PcNrdD.

| Ligands | dATP tetramer | dATP dimer | ATP–dTTP–GTP dimer | ATP–dGTP dimer | ATP–CTP dimer | dATP–CTP tetramer | dATP–CTP dimer |
|---|---|---|---|---|---|---|---|
| PDB entry | 8P28 | 8P27 | 8P2S | 8P39 | 8P23 | 8P2C | 8P2D |
| EMDB entry | EMD-17359 | EMD-17358 | EMD-17373 | EMD-17385 | EMD-17357 | EMD-17360 | EMD-17361 |
| Concentrations (mM) | | 0.5 | 5.0/1.0 / 1.0 | 5.0/1.0 | 0.5/0.5 | | 0.5/0.5 |
| Grids | | Quantifoil 1.2/1.3, 300 mesh Cu | Quantifoil 2.1, 300 mesh Au | Quantifoil 2.1, 300 mesh Au | Quantifoil 2.1, 300 mesh Cu | | Quantifoil 2.1, 300 mesh Cu |
| Pixel size (Å) | | 0.8676 | 0.824 | 0.860 | 0.8676 | | 0.8464 |
| Dose rate (e⁻/px/s) | | 16.1 | 15.0 | 15.0 | 17.1 | | 13.7 |
| Exposure time (s) | | 2.0 | 2.0 | 2.0 | 2.1 | | 2.0 |
| Total dose (e⁻/Å²) | | 46 | 40 | 40 | 48 | | 38 |
| Defocus range (µM) | | −1.0 to −3.4 | −0.8 to −2.2 | −0.6 to −2.4 | −0.6 to −2.4 | | −0.6 to −2.0 |
| Micrographs used (collected) | | 16,667 (21,804) | 4964 (12,501) | 14,346 (16,745) | 17,033 (21,512) | 11,780 (15,268) | 8796 (15,268) |
| Particles in final class | 1,349,133 | 1,009,021 | 437,866 | 589,345 | 291,231 | 1,105,348 | 1,132,695 |
| Symmetry | C2 | C1 | C1 | C1 | C1 | C2 | C1 |
| Resolution (FSC = 0.143; Å) | 2.77 | 2.73 | 2.40 | 2.56 | 3.17 | 2.59 | 2.59 |
| Map sharpening B-factor (Å²) | 138.5 | 128.0 | 70.5 | 72.3 | 152.8 | 107.8 | 108.9 |
| CC (mask) from phenix.refine | 0.76 | 0.77 | 0.76 | 0.63 | 0.74 | 0.81 | 0.64 |
| Model composition | | | | | | | |
| Non-hydrogen atoms (residues) | 21,141 (2584) | 8710 (1089) | 9694 (1201) | 9705 (1200) | 11,399 (1401) | 21,137 (2584) | 8793 (1087) |
| Ligands | 8 dATP (a) 4 dATP (s) | 2 dATP (s) | 2 dTTP (s) 1 GTP (c) | 2 dGTP (s) 1 ATP (c) | 4 ATP (a) 1 CTP (c) | 8 dATP (a) 4 dATP (s) | 2 dATP (s) |
| min/mean/max B-factors protein (Å²) | 52.8/63.4/84.7 | 40.2/50.6/66.6 | 25.6/35.4/72.9 | 12.6/41.2/98.7 | 15.7/66.1/124.4 | 15.5/49.4/136.4 | 30.9/43.6/78.1 |
| min/mean/max B-factors ligands (Å²) | 55.8/62.9/74.5 | 47.1/47.2/47.4 | 25.4/33.1/79.3 | 10.8/33.1/88.0 | 26.0/81.2/123.7 | 18.0/46.8/92.7 | 35.9/36.2/47.2 |
| Deviations from ideal geometry | | | | | | | |
| rmsd (bonds) | 0.005 | 0.004 | 0.003 | 0.003 | 0.004 | 0.004 | 0.002 |
| rmsd (angles) | 0.61 | 0.55 | 0.54 | 0.60 | 0.58 | 0.59 | 0.51 |
| Ramachandran plot (%) | | | | | | | |
| Favoured | 90.1 | 91.7 | 95.2 | 93.8 | 93.8 | 94.1 | 92.0 |
| Allowed | 9.9 | 8.3 | 4.7 | 5.8 | 5.9 | 5.9 | 8.0 |
| Outliers | 0.0 | 0.0 | 0.1 | 0.4 | 0.3 | 0.0 | 0.0 |
| Rotamer outliers | 1.9 | 5.4 | 3.0 | 6.0 | 3.8 | 4.2 | 4.8 |
| MolProbity score | 2.1 | 2.4 | 2.2 | 2.5 | 2.2 | 2.3 | 2.5 |
| MolProbity clash score | 6.5 | 6.2 | 8.7 | 10.0 | 5.5 | 7.1 | 10.2 |

Grids were clipped and loaded into a 300-kV Titan Krios G2 microscope (Thermo Fisher Scientific, EPU 2.8.1 software) equipped with a Gatan BioQuantum energy filter and a K3 Summit direct electron detector (AMETEK). Grids were screened for quality based on particle distribution and density, and images from the best grid were recorded. Micrographs were recorded at a nominal magnification of 105,000. Details of the other data collection parameters used for each sample are given in *Table 5*.

## Cryo-EM data processing

Data processing was performed in cryoSPARC (*Punjani et al., 2017*). The first steps in processing of all datasets were patch motion correction and patch contrast transfer function (CTF) estimation, followed by curation of exposures based on ice thickness, poor defocus estimation, etc. Other steps are as detailed below. Resolution estimation in cryoSPARC was done using a gold standard Fourier shell correlation value of 0.143 (*Rosenthal and Henderson, 2003*). Map post-processing was done using DeepEMhancer (*Sanchez-Garcia et al., 2021*) and post-processing resolution estimation was performed using DeepRes (*Ramírez-Aportela et al., 2019*). Models were placed in the maps using either phenix.dock_in_map (*Liebschner et al., 2019*) or Molrep (*Vagin and Teplyakov, 1997*) in the CCP-EM package (*Burnley et al., 2017*). Model building was done by alternating rounds of manual building in Coot (*Emsley et al., 2010*) with real space refinement in phenix.refine (*Liebschner et al., 2019*). Secondary structure restraints were used throughout. The inclusion of riding hydrogen atoms reduced the number of bad contacts in all models. An AlphaFold2 (*Jumper et al., 2021*) model of PcNrdD was used in the later stages of building the first model to resolve some ambiguous loops. All final models have good correlations between map and model and good stereochemical properties (*Table 5*). All structures have been deposited in the Protein Databank and the corresponding volumes deposited in the Electron Microscopy Data Bank with the accession numbers listed in *Table 5*.

### PcNrdD–ATP–CTP complex
#### Data processing
A total of 17,033 movies were used after curation. A low-pass filtered volume was made in Chimera (*Pettersen et al., 2021*) from the crystal structure of NrdD from *T. maritima* (TmNrdD, PDB ID 4COI). From this volume, a set of templates was created and used for template-based picking with a diameter of 100 Å. Due to the large number of micrographs, they were split into two sets. The largest set (16,219 micrographs) was analysed first. About 8.6 million particles were extracted with a 350-pixel box size and classified into 60 2D classes. Eight representative classes containing different orientations (~3.3 million particles) were used to generate four ab initio models without symmetry (C1) and particles were subjected to 3D heterogeneous refinement using four classes. The most populated model containing ~1 million particles was subjected to homogeneous and non-uniform refinement, which gave a map with an overall FSC resolution of 2.94 Å. A further 435,695 particles were then extracted from the remaining 810 micrographs. These were subjected to 2D classification, and the resulting classes were merged with the larger set, giving 1,199,575 particles. The combined particles were subjected to homogeneous, non-uniform and local refinement with a predefined mask, which gave a map at 2.87 Å (*Figure 6—figure supplement 1*).

#### Generation of map with better ATP-cone density
The particles were further classified into 50 2D classes, from which 23 classes having 1,040,266 particles were selected. Four ab initio models were generated and 570,730 particles with slightly better ATP-cone density were refined without symmetry. The particles were then subjected to 3D classification using ten classes of ~57,000 particles each. The classes having slightly better ATP-cone density were selected for further processing and final refinement without symmetry gave a map with an overall FSC resolution of 3.17 Å from 291,231 particles, which was post-processed using DeepEMhancer (*Figure 6—figure supplement 1*).

#### Model building and refinement
A partially complete model for one dimer of the dATP complex (see below) was placed in the map using phenix.dock_in_map. The ATP-cones were built based on the ones from the dATP-only tetramers (see below).

## PcNrdD–dATP complexes

### Tetramers

#### Data processing for first part

A set of 11,161 movies was used after curation, template-based picking based on the structure of TmNrdD and particle extraction using a box size of 448 pixels gave 5,331,420 particles. Seven 2D classes having 1,171,839 particles were used to make ab initio models, followed by heterogeneous refinement. The best class having 461,020 particles and volume was used for homogeneous refinement, which gave a map at 3.3 Å. Non-uniform refinement followed by local refinement with a predetermined mask gave a map at 3.1 Å with no imposed symmetry.

#### Data processing for second part

A set of 16,667 movies was used after curation. Particles were picked and extracted using the same box size as in the first part, which gave 7,889,234 particles. Ten 2D classes having 1,881,057 particles were used to make two ab initio models, followed by heterogeneous refinement. The best class with 1,083,657 particles was selected and subjected to homogeneous refinement followed by non-uniform refinement and a local new refinement with a pre-determined mask, which gave a map at 2.9 Å with no imposed symmetry.

#### Merging of datasets and CTF refinement

The best particles from the first dataset were classified into 80 2D classes from which 36 having 399,523 particles were selected. The particles from the second dataset resulting in the best volume were classified into 100 2D classes, of which 31 having 949,610 particles were selected. These particle sets were merged (giving 1,349,133 particles) and refined against the best 3D volume which gave a map at 2.78 Å. Iterative refinement and NU refinement jobs, ultimately using C2 symmetry, and local CTF refinement with a defocus search range of ±2000, gave a map at 2.8 Å with C2 symmetry that was post-processed using DeepEMhancer (*Figure 9—figure supplement 1*).

#### Model building and refinement

Model building was initially done for one dimer by fitting a homology model generated using Swiss-Model with T4NrdD as a template to the map using phenix.dock_in_map. Similar homology models for the four ATP-cones were placed by hand into the density followed by rounds of real space refinement. After almost completely rebuilding one dimer, the second dimer was placed into the density and model building and refinement were continued.

### Dimers

The first part of the data processing was shared with the PcNrdD–dATP tetramers. In the same micrographs there were a number of 2D class averages that represented dimers. Eight such class averages having 1,916,387 particles were used to make two ab initio models. This was followed by rounds of heterogeneous refinement from where the best model having 1,009,021 particles was refined with no symmetry to give a map at 2.8 Å. A final non-uniform refinement with C1 symmetry gave a map at 2.6 Å that was post-processed using DeepEMhancer (*Figure 9—figure supplement 1*).

## PcNrdD–dATP–CTP complexes

### Tetramers

Blob picking was carried out from 11,780 curated micrographs with a maximum diameter of 200 Å and a minimum diameter of 90. In total 4,646,479 particles were then extracted with a box size of 448 pixels. This was followed by 2D classification, ab initio model generation and heterogeneous refinements. The best class with 1,105,348 particles was used for non-uniform refinement with C2 symmetry, which gave a final map at 2.59 Å resolution that was post-processed using DeepEMhancer (*Figure 9—figure supplement 5*).

### Dimers

The initial data processing steps were the same as above. Template-based picking from 8796 curated micrographs using a low-pass filtered volume of the dATP-only tetramer gave 6,836,668 particles after extraction with a box size of 300 pixels. This was followed by 2D classifications, selection of classes

representing dimers, ab initio model generations and refinement of the best class. The final round of non-uniform refinement with 1,132,695 particles with C1 symmetry gave a map of 2.53 Å resolution. The estimated median resolution after post-processing was 2.1 Å (*Figure 9—figure supplement 5*). Model building used the coordinates of the dATP-only dimer as a starting model.

### PcNrdD–ATP–dTTP–GTP complex

### Data processing

A total of 12,501 movies were processed and 6105 movies were used after curation. Template-based picking was used with a diameter of 150 pixels. About 3.9 million particles were extracted from 5952 movies with a 400 pixel box size and used for 2D classification into 100 classes. Eleven representative 2D classes containing different orientations (~2.4 million particles) were used to generate an ab initio model and particles were subjected to 3D heterogeneous refinement using ten classes, resulting in the most populated model containing 437,866 particles. This model was subjected to homogeneous and non-uniform refinement, which gave a map with an overall FSC resolution of 2.47 Å. The particles were then subjected to 3D classification and the resulting classes were merged for further processing and final refinement without symmetry (C1) gave a map with an overall FSC resolution of 2.4 Å that was post-processed using DeepEMhancer (*Figure 8—figure supplement 1*).

### Model building and refinement:

A complete model for the dimer of the ATP–CTP structure (see above) was placed in the map using phenix.dock_in_map. The ATP-cones were removed as they were not visible in the reconstruction. After model building and refinement, the final map-to-model correlation value was 0.74. Model quality statistics are presented in *Table 5*.

### PcNrdD–ATP–dGTP complex

### Data processing

Template-based picking was carried out on 14,373 curated micrographs and 4,522,028 particles were extracted with a box size of 350 pixels. This was followed by multiple rounds of 2D classifications, ab initio model generation and refinements. C1 symmetry was applied on the best volume which gave a map with 589,345 particles having a final resolution of 2.58 Å. Model building and refinement used the PcNrdD–ATP/CTP complex as a starting model (*Figure 8—figure supplement 5*).

## Bioinformatics

To construct an HMM logo of NrdD sequences, a representative selection of NrdD sequences from all domains of life including viruses were collected and aligned with Clustal Omega (*Sievers et al., 2011*). Subsequently, an HMM model was built with hmmbuild from the HMMER suite (*Eddy, 2011*) and a logo was created using the Skylign web service using the 'Information Content - Above Background' option (*Wheeler et al., 2014*). Conserved parts of the logo were extracted and displayed here.

## Hydrogen–deuterium exchange mass spectrometry

All chemicals were from Sigma-Aldrich. pH measurements were made using a SevenCompact pH-meter equipped with an InLab Micro electrode (Mettler-Toledo). A 4-point calibration (pH 2, 4, 7, 10) was made prior to all measurements. The HDX-MS analysis was made using automated sample preparation on a LEAP H/D-X PAL platform interfaced to an liquid chromatography-mass spectrometry (LC–MS) system, comprising an Ultimate 3000 micro-LC coupled to an Orbitrap Q Exactive Plus MS.

HDX was performed on 2 mg/ml PcNrdD with and without ligands (5 mM ATP + 5 mM CTP or 1 mM dATP + 5 mM CTP) in 25 mM HEPES, 100 mM KCl, 20 mM MgCl$_2$, and 0.5 mM TCEP, pH 7.5. The HDX-MS was performed in one continuous run, the apo state being run before both complexed states. For each labelling time point, 3 μl HDX samples were diluted with 27 μl labelling buffer (containing the ligands) of the same composition prepared in D$_2$O, pH$_{(read)}$ 7.4. The HDX labelling was carried out at $t$ = 0, 30, 300, and 3000 s at 20°C. Each time point was run in four to six replicates. The labelling reaction was quenched by dilution of 30 μl labelled sample with 30 μl of 1% trifluoroacetic acid, 0.4 M TCEP, 4 M urea, pH 2.5 at 1°C. 50 μl of the quenched sample was directly injected and

subjected to online pepsin digestion at 4°C on an in-house immobilised pepsin column (2.1 × 30 mm). The online digestion and trapping were performed for 4 min using a flow of 50 μl/min 0.1% FA, pH 2.5. The peptides generated by pepsin digestion were subjected to online solid phase extraction (SPE) on a PepMap300 C18 trap column (1 × 15 mm) and washed with 0.1% FA for 60 s. Thereafter, the trap column was switched in-line with a reversed-phase analytical column (Hypersil GOLD, particle size 1.9 μm, 1 × 50 mm). The mobile phases were (A) 0.1% FA and 95% acetonitrile/0.1% FA (B) and separation was performed at 1°C using a gradient of 5–50% B over 8 min and then from 50 to 90% B for 5 min. Following the separation, the trap and column were equilibrated at 5% organic content until the next injection. The needle port and sample loop were cleaned three times after each injection with mobile phase 5% MeOH/0.1% FA, followed by 90% MeOH/0.1% FA and a final wash of 5% MeOH/0.1% FA. After each sample and blank injection, the pepsin column was washed by injecting 90 μl of pepsin wash solution 1% FA/4 M urea/5% MeOH. In order to minimise carry-over, a full blank was run between each sample injection. Separated peptides were analysed on a Q Exactive Plus MS, equipped with a HESI source operated at a capillary temperature of 250°C with sheath gas 12, Aux gas 2, and sweep gas 1 (au). For HDX analysis, MS full scan spectra were acquired at 70 K resolution, AGC 3e6, max IT 200 ms and scan range 300–2000. For identification of generated peptides, separate non-deuterated samples were analysed using data-dependent MS/MS with HCD fragmentation. A summary of the HDX experimental detail is reported in *Figure 10—source data 1*. The mass spectrometry raw files have been deposited at the ProteomeExchange Consortium via the PRIDE partner repository (*Perez-Riverol et al., 2022*) with the dataset identifier PXD047943.

## Data analysis

PEAKS Studio X Bioinformatics Solutions Inc (BSI, Waterloo, Canada) was used for peptide identification after pepsin digestion of non-deuterated samples. The search was done on a FASTA file with only the NrdD sequence. The search criteria were a mass error tolerance of 15 ppm and a fragment mass error tolerance of 0.05 Da, allowing for fully unspecific cleavage by pepsin. Peptides identified by PEAKS with a peptide score value of log p > 25 and no modifications were used to generate a peptide list containing peptide sequence, charge state, and retention time for the HDX analysis. HDX data analysis and visualisation were performed using HDExaminer, version 3.3 (Sierra Analytics Inc, Modesto, US). The analysis was made on the best charge state for each peptide, allowed only for EX2 (except for bimodal parts of the protein for which EX1 calculation of uptake was allowed) and the two first residues of a peptide were assumed unable to hold deuteration. Due to the comparative nature of the measurements, the deuterium incorporation levels for the peptic peptides were derived from the observed relative mass difference between the deuterated and non-deuterated peptides without back-exchange correction using a fully deuterated sample (*Engen and Wales, 2015*). As a full deuteration experiment was not made, full deuteration was set to 75% of maximum theoretical uptake. The presented deuteration data are the average of all high and medium confidence results. The allowed retention time window was ±0.5 min. The spectra for all time points were manually inspected; low scoring peptides, obvious outliers, and any peptides where retention time correction could not be made consistent were removed. As bottom-up labelling HDX-MS is limited in structural resolution by the degree of overlap of the peptides generated by pepsin digestion, the peptide map overlap is shown for the respective state in *Figure 10—source data 1*.

## Materials availability statement

The plasmids for expression of PcNrdD and PcNrdG are available from the authors upon request.

## Acknowledgements

We dedicate this article to PhD candidate Ipsita Banerjee[¶] who made seminal contributions to the data processing parts of this study but who passed away suddenly on 7 December 2022. The authors thank Gustav Berggren, Uppsala University, Martin Högbom, Stockholm University, and Marc Fontecave, Collège de France, for letting us use their anaerobic chambers, Anders Hofer, Umeå University, for the GEMMA instrument, David Drew and Henrietta Nielsen, Stockholm University, for the ITC and MST instruments, Anders Olsson, SciLifeLab Stockholm University, for the ITC and HPLC instruments, Annette Roos, SciLifeLab Uppsala University, for her help during the procedure of PcNrdD labelling for MST, Alexey Pisarev and Thomas Jonsen, Agilent, for fixing our HPLC instrument, and Malvern

Panalytical for kindly sharing the MicroCal PEAQ-ITC analysis software for the analysis of ITC data. We would like to also thank master student Sina Becker for SEC experiments. Cryo-EM sample screening, optimisation, and data collection were performed at the Cryo-EM Swedish National Facility, funded by the Knut and Alice Wallenberg, Family Erling Persson and Kempe Foundations, SciLifeLab, Stockholm University, and Umeå University. The authors would like to thank Marta Carroni, Karin Walldén, Julian Conrad, Terezia Kovalova, and Victor Tobiasson for their assistance during the cryo-EM experiments. Support from the Swedish National Infrastructure for Biological Mass Spectrometry (BioMS) and the SciLifeLab, Integrated Structural Biology platform is gratefully acknowledged.

This study was supported by grants from the Swedish Research Council (2019-01400 to BMS, 2016-04855 to DTL), the Swedish Cancer Society (CAN 20 1210 PjF to BMS), and the Wenner-Gren Foundations (to BMS).

## Additional information

### Funding

| Funder | Grant reference number | Author |
|---|---|---|
| Vetenskapsrådet | 2019-01400 | Britt-Marie Sjöberg |
| Vetenskapsrådet | 2016-04855 | Derek T Logan |
| Cancerfonden | CAN 20 1210 PjF | Britt-Marie Sjöberg |
| Wenner-Gren Stiftelserna | | Britt-Marie Sjöberg |

The funders had no role in study design, data collection, and interpretation, or the decision to submit the work for publication.

### Author contributions

Ornella Bimai, Conceptualization, Data curation, Formal analysis, Investigation, Writing – review and editing, Visualization; Ipsita Banerjee, Lucas Hultgren, Data curation, Formal analysis, Investigation; Inna Rozman Grinberg, Data curation, Formal analysis, Investigation, Writing – original draft, Writing – review and editing, Visualization; Ping Huang, Data curation, Investigation; Simon Ekström, Data curation, Formal analysis, Validation, Investigation, Visualization, Writing – review and editing; Daniel Lundin, Data curation, Formal analysis, Validation, Investigation, Visualization; Britt-Marie Sjöberg, Conceptualization, Resources, Data curation, Formal analysis, Supervision, Funding acquisition, Validation, Writing – original draft, Project administration, Writing – review and editing, Investigation, Visualization; Derek T Logan, Conceptualization, Resources, Data curation, Formal analysis, Supervision, Funding acquisition, Validation, Investigation, Visualization, Writing – original draft, Project administration, Writing – review and editing

### Author ORCIDs

Ornella Bimai http://orcid.org/0000-0003-0562-7251
Inna Rozman Grinberg https://orcid.org/0000-0003-3094-1998
Ping Huang https://orcid.org/0000-0002-7676-6905
Simon Ekström http://orcid.org/0000-0002-7694-285X
Daniel Lundin https://orcid.org/0000-0002-8779-6464
Britt-Marie Sjöberg https://orcid.org/0000-0001-5953-3360
Derek T Logan https://orcid.org/0000-0002-0098-8560

Reviewer #3 (Public Review): https://doi.org/10.7554/eLife.89292.4.sa1
Author response https://doi.org/10.7554/eLife.89292.4.sa2

# Additional files

## Supplementary files
• MDAR checklist

## Data availability
All structures have been deposited in the Protein Data Bank and the corresponding volumes deposited in the Electron Microscopy Data Bank with the accession numbers listed below.

The following datasets were generated:

| Author(s) | Year | Dataset title | Dataset URL | Database and Identifier |
|---|---|---|---|---|
| Banerjee I, Bimai O, Sjöberg BM, Logan DT | 2023 | Cryo-EM structure of the anaerobic ribonucleotide reductase from Prevotella copri in its dimeric, dATP-bound state | https://doi.org/10.2210/pdb8P27/pdb | Worldwide Protein Data Bank, 10.2210/pdb8P27/pdb |
| Banerjee I, Bimai O, Sjöberg BM, Logan DT | 2023 | Cryo-EM structure of the anaerobic ribonucleotide reductase from Prevotella copri in its tetrameric, dATP-bound state | https://doi.org/10.2210/pdb8P28/pdb | Worldwide Protein Data Bank, 10.2210/pdb8P28/pdb |
| Bimai O, Banerjee I, Sjöberg BM, Logan DT | 2023 | Cryo-EM structure of the anaerobic ribonucleotide reductase from Prevotella copri in its dimeric, ATP/dTTP/GTP-bound state | https://doi.org/10.2210/pdb8P2S/pdb | Worldwide Protein Data Bank, 10.2210/pdb8P2S/pdb |
| Bimai O, Banerjee I, Sjöberg BM, Logan DT | 2023 | Cryo-EM structure of the anaerobic ribonucleotide reductase from Prevotella copri in its dimeric, dGTP/ATP-bound state | https://doi.org/10.2210/pdb8P39/pdb | Worldwide Protein Data Bank, 10.2210/pdb8P39/pdb |
| Banerjee I, Bimai O, Sjöberg BM, Logan DT | 2023 | Cryo-EM structure of the anaerobic ribonucleotide reductase from Prevotella copri in its dimeric, ATP/CTP-bound state | https://doi.org/10.2210/pdb8P23/pdb | Worldwide Protein Data Bank, 10.2210/pdb8P23/pdb |
| Banerjee I, Bimai O, Sjöberg BM, Logan DT | 2023 | Cryo-EM structure of the anaerobic ribonucleotide reductase from Prevotella copri in its tetrameric state produced in the presence of dATP and CTP | https://doi.org/10.2210/pdb8P2C/pdb | Worldwide Protein Data Bank, 10.2210/pdb8P2C/pdb |
| Banerjee I, Bimai O, Sjöberg BM, Logan DT | 2023 | Cryo-EM structure of the dimeric form of the anaerobic ribonucleotide reductase from Prevotella copri produced in the presence of dATP and CTP | https://doi.org/10.2210/pdb8P2D/pdb | Worldwide Protein Data Bank, 10.2210/pdb8P2D/pdb |
| Banerjee I, Bimai O, Sjöberg BM, Logan DT | 2023 | Cryo-EM structure of the anaerobic ribonucleotide reductase from Prevotella copri in its tetrameric, dATP-bound state | https://www.ebi.ac.uk/emdb/EMD-17359 | EMDB Electron Microscopy Data Bank, EMD-17359 |
| Banerjee I, Bimai O, Sjöberg BM, Logan DT | 2023 | Cryo-EM structure of the anaerobic ribonucleotide reductase from Prevotella copri in its dimeric, dATP-bound state | https://www.ebi.ac.uk/emdb/EMD-17358 | EMDB Electron Microscopy Data Bank, EMD-17358 |

*Continued on next page*

*Continued*

| Author(s) | Year | Dataset title | Dataset URL | Database and Identifier |
|---|---|---|---|---|
| Bimai O, Banerjee I, Sjöberg BM, Logan DT | 2023 | Cryo-EM structure of the anaerobic ribonucleotide reductase from Prevotella copri in its dimeric, ATP/dTTP/GTP-bound state | https://www.ebi.ac.uk/emdb/EMD-17373 | EMDB Electron Microscopy Data Bank, EMD-17373 |
| Bimai O, Banerjee I, Logan DT | 2023 | Cryo-EM structure of the anaerobic ribonucleotide reductase from Prevotella copri in its dimeric, dGTP/ATP-bound state | https://www.ebi.ac.uk/emdb/EMD-17385 | EMDB Electron Microscopy Data Bank, EMD-17385 |
| Banerjee I, Bimai O, Sjöberg BM, Logan DT | 2023 | Cryo-EM structure of the anaerobic ribonucleotide reductase from Prevotella copri in its dimeric, ATP/CTP-bound state | https://www.ebi.ac.uk/emdb/EMD-17357 | EMDB Electron Microscopy Data Bank, EMD-17357 |
| Banerjee I, Bimai O, Sjöberg BM, Logan DT | 2023 | Cryo-EM structure of the anaerobic ribonucleotide reductase from Prevotella copri in its tetrameric state produced in the presence of dATP and CTP | https://www.ebi.ac.uk/emdb/EMD-17360 | EMDB Electron Microscopy Data Bank, EMD-17360 |
| Banerjee I, Bimai O, Sjöberg BM, Logan DT | 2023 | Cryo-EM structure of the dimeric form of the anaerobic ribonucleotide reductase from Prevotella copri produced in the presence of dATP and CTP | https://www.ebi.ac.uk/emdb/EMD-17361 | EMDB Electron Microscopy Data Bank, EMD-17361 |

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
