## [Editor Report · eLife assessment]

This study advances our understanding of the allosteric regulation of anaerobic ribonucleotide reductases (RNRs) by nucleotides, providing **valuable** new structural insight into class III RNRs containing ATP cones. The cryo-EM structural characterization of the system is **solid**, but some open questions remain about the interpretation of activity/binding assays and the HDX-MS results that have been newly incorporated compared to a previous version. The work will be of interest to biochemists and structural biologists working on ribonucleotide reductases and other allosterically regulated enzymes.

---

## [Referee Report · Reviewer #3 (Public Review)]

The manuscript by Bimai et al describes a structural and functional characterization of an anaerobic ribonucleotide reductase (RNR) enzyme from the human microbe, P. copri. More specifically, the authors aimed to characterize the mechanism by how (d)ATP modulates nucleotide reduction in this anaerobic RNR, using a combination of enzyme kinetics, binding thermodynamics, and cryo-EM structural determination, complemented by hydrogen-deuterium exchange (HDX). One of the principal findings of this paper is the ordering of a NxN 'flap' in the presence of ATP that promotes RNR catalysis and the disordering (or increased protein dynamics) of both this flap and the glycyl radical domain (GRD) when the inhibitory effector, dATP, binds. The latter is correlated with a loss of substrate binding, which is the likely mechanism for dATP inhibition. It is important to note that the GRD is remote (>30 Ang) from the binding site of the dATP molecule, suggesting long-range communication of the structural (dis)ordering. The authors also present evidence for a shift in oligomerization in the presence of dATP. The work does provide evidence for new insights/views into the subtle differences of nucleotide modulation (allostery) of RNR, in a class III system, through long-range interactions.

The strengths of the work are the impressive, in-depth structural analysis of the various regulated forms of PcRNR by (d)ATP using cryo-EM. The authors present seven different models in total, with striking differences in oligomerization and (dis)ordering of select structural features, including the GRD that is integral to catalysis. The authors present several, complementary biochemical experiments (ITC, MST, EPR, kinetics) aimed at resolving the binding and regulatory mechanism of the enzyme by various nucleotides. The authors present a good breadth of the literature in which the focus of allosteric regulation of RNRs has been on the aerobic orthologues.

The addition of hydrogen-deuterium exchange mass spectrometry (HDX-MS) complements the results originating from cryo-EM data. Most notable, is the observation of the enhanced exchange (albeit quite subtle) of the GRD domain in the presence of dATP that matches the loss of structural information in this region in the cryo-EM data. The most pronounced and compelling HDX results are seen in the form of dATP-induced protection of peptides immediately adjacent to the b-hairpin at the s-site, where dATP is expected to bind based on cryo-EM. It is clear that the presence of dATP increases the rigidity of this region.

---

## [Author Response]

The following is the authors’ response to the previous reviews.

**eLife assessment**
This study advances our understanding of the allosteric regulation of anaerobic ribonucleotide reductases (RNRs) by nucleotides, providing valuable new structural insight into class III RNRs containing ATP cones. The cryo-EM structural characterization of the system is solid, but some open questions remain about the interpretation of activity/binding assays and the newly incorporated HDX-MS results. The work will be of interest to biochemists and structural biologists working on ribonucleotide reductases and other allosterically regulated enzymes.
**Public Reviews:**

**Reviewer #1 (Public Review):**
The goal of this study is to understand the allosteric mechanism of overall activity regulation in an anaerobic ribonucleotide reductase (RNR) that contains an ATP-cone domain. Through cryo-EM structural analysis of various nucleotide-bound states of the RNR, the mechanism of dATP inhibition is found to involve order-disorder transitions in the active site. These effects appear to prevent binding of substrate and a radical transfer needed to initiate the reaction.Strengths of the manuscript include the comprehensive nature of the work - including both numerous structures of different forms of the RNR and detailed characterization of enzyme activity to establish the parameters of dATP inhibition. The manuscript has been improved in a revision by performing additional experiments to help corroborate certain aspects of the study. But these new experiments do not address all of the open questions about the structural basis for mechanism. Additionally, some questions about the strength of biochemical data and fit of binding or kinetic curves to data that were raised by other referees still remain. Some experimental observations are not consistent with the proposed model. For example, why does dATP enhance Gly radical formation when the proposed mechanism of dATP inhibition involves disorder in the Gly radical domain?The work is impactful because it reports initial observations about a potentially new mode of allosteric inhibition in this enzyme class. It also sets the stage for future work to understand the molecular basis for this phenomenon in more detail.

We express our gratitude to the reviewer for dedicating time to review our work and for the overall favorable assessment. We agree that the question of exactly how much the glycyl radical domain becomes more mobile without losing the glycyl radical entirely is an unresolved one but we also think that our work sets a solid basis for future experiments by us and others.

**Reviewer #3 (Public Review):**
The manuscript by Bimai et al describes a structural and functional characterization of an anaerobic ribonucleotide reductase (RNR) enzyme from the human microbe, P. copri. More specifically, the authors aimed to characterize the mechanism by how (d)ATP modulates nucleotide reduction in this anaerobic RNR, using a combination of enzyme kinetics, binding thermodynamics, and cryo-EM structural determination, complemented by hydrogen-deuterium exchange (HDX). One of the principal findings of this paper is the ordering of a NxN 'flap' in the presence of ATP that promotes RNR catalysis and the disordering (or increased protein dynamics) of both this flap and the glycyl radical domain (GRD) when the inhibitory effector, dATP, binds. The latter is correlated with a loss of substrate binding, which is the likely mechanism for dATP inhibition. It is important to note that the GRD is remote (>30 Ang) from the binding site of the dATP molecule, suggesting long-range communication of the structural (dis)ordering. The authors also present evidence for a shift in oligomerization in the presence of dATP. The work does provide evidence for new insights/views into the subtle differences of nucleotide modulation (allostery) of RNR, in a class III system, through long-range interactions.The strengths of the work are the impressive, in-depth structural analysis of the various regulated forms of PcRNR by (d)ATP using cryo-EM. The authors present seven different models in total, with striking differences in oligomerization and (dis)ordering of select structural features, including the GRD that is integral to catalysis. The authors present several, complementary biochemical experiments (ITC, MST, EPR, kinetics) aimed at resolving the binding and regulatory mechanism of the enzyme by various nucleotides. The authors present a good breadth of the literature in which the focus of allosteric regulation of RNRs has been on the aerobic orthologues.The addition of hydrogen-deuterium exchange mass spectrometry (HDX-MS) complements the results originating from cryo-EM data. Most notably, is the observation of the enhanced exchange (albeit quite subtle) of the GRD domain in the presence of dATP that matches the loss of structural information in this region in the cryo-EM data. The most pronounced and compelling HDX results are seen in the form of dATP-induced protection of peptides immediately adjacent to the b-hairpin at the s-site, where dATP is expected to bind based on cryo-EM. It is clear that the presence of dATP increases the rigidity of this region.

We are happy that both reviewers find the HDX-MS experiments to be a valuable addition to the existing data.

Weaknesses:The discussion of the change in peptide mobility in the N-terminal region is complicated by the presence of bimodal mass spectral features and this may prevent detailed interpretation of the data, especially for select peptide region that shows opposite trends upon nucleotide association.Further, the HDX data in the NxN flap is unchanged upon nucleotide binding (ATP, dATP, or CTP), despite changes observed in the cryo-EM data.

We are grateful to the reviewer for the comprehensive feedback on the HDX-MS part and for identifying areas for improvement. The HDX analysis was of course undertaken with the intention of identifying differences in disorder of the NxN flap and GRD region. From an HDX perspective both regions were found to be highly susceptible to HDX regardless of state/ligand, due to surface accessibility and/or very fast dynamics. However, this does not mean that there is no difference in the degree of order of these regions upon ligand addition, simply that we with HDX-MS, in the limited time span of 30-3000 seconds, could not conclusively support an increased disorder. We have rephrased the discussion text to reflect this fact

**Recommendations for the authors:**

**Reviewer #1 (Recommendations For The Authors):**
On page 5 (and throughout the manuscript) there are some inconsistencies in how dissociation constants for effectors and inhibitors are described - for example, D in KD is sometimes subscripted and sometimes not.

Thank you for noticing these remaining errors. We hope that we have fixed all of them now.

**Reviewer #3 (Recommendations For The Authors):**
The authors addressed many of the initial concerns raised. The addition of the HDX-MS data in this revision is a welcomed contribution to the work and complements the cryo-EM data. In select cases, the data may be over-interpreted. This reviewer suggests that the authors revise the text in this section so that it is more consistent with the presented data.Specific points:(1) The bimodal mass spectral features in the N-terminal domain complicate the data interpretation. Specifically for peptides in 81-99 region, the fast exchanging feature shows protection in the presence of (d)ATP/CTP, but the opposite trend is observed for the slow exchanging species. It is therefore advisable to not make absolutes about the HDX results in this region, as the data are complicated.

As stated by the reviewer, it is not possible from the presented HDX data to deduce if this is a result of 50% loaded dimer or the oligomerization state of the protein. We have remedied this by removing mentions of a difference between the dATP and ATP in bimodality. Also, we have addressed this in the text by stating that the main reason is most likely the different oligomerization states present in solution. Nevertheless, it is clear from the HDX data that the N-terminal region and 81-99 are very interesting, and it was somewhat disappointing that due to the dynamics of the oligomerization it was not possible to SEC-purify pure dimer or tetramer samples for HDX-MS, in order to deconvolute the cause.

(2) Related to #1, the authors assign the bimodal HDX behavior to EX1 mechanism, but this is not necessarily (and unlikely) true based on the limited time points. The authors also state that it originates from the heterogeneity of the sample: "a mixture of states" which could reflect the mixture of oligomerization states. The authors should be careful assigning EX1 mechanism unless there are compelling results to support it.

We apologize for the unfortunate phrasing. It was not our intention to imply that the bimodality is due to true EX1 kinetics. See the above answer. The mention of EX1 has been removed from the discussion text.

(3) The deuterium uptake for peptide 118-126 is very small (~1Da) compared to the length of the peptide. The change in deuterium uptake (<0.25Da) from dATP is very small; the authors should proceed with caution when presenting interpretations of such small differences.

We agree with the reviewer that extra caution should be taken when dealing with such a small difference. However, the 118-126 peptide has been significance tested in both HDExaminer and Deuteros 2.0, and we also observed this for more than one run. The difference in uptake is small but increases to significance at the longer labelling times. The proximity to the NxN flap makes it interesting in context of an allosteric conformational change. i.e the dynamics of the NxN might be too fast so we can only see some secondary effects. We would like to keep the data in Figure 10 for reasons of transparency. In essence this is similar to the observed bimodality mentioned above: we cannot fully explain the observation but present the data as it was observed.

(4) On p. 22, the authors should consider revising the following statement: "confirming dATP binding to the s-site." Even though the HDX data are most compelling for the protection of peptides 178-204 and 330-348 that are adjacent to the beta-hairpin at the s-site, these data cannot "confirm" a binding site for a small molecule, such as dATP.

We appreciate that the reviewer has pointed out that the statement can be misleading, and we agree that the binding site of small molecules can’t be confirmed based solely on HDX data. The sentence reformulated to clarify that the binding site was confirmed based on the combined evidence of HDX data and the previously presented biochemical and structural data on the s-site.